# Kernel-based Equalized Odds: A Quantification of Accuracy-Fairness Trade-off in Fair Representation Learning

**Yijin Ni**
H. Milton Stewart School of Industrial
and Systems Engineering
Georgia Institute of Technology
`yni64@gatech.eu`

**Xiaoming Huo**
H. Milton Stewart School of Industrial
and Systems Engineering
Georgia Institute of Technology
`huo@gatech.eu`

## Abstract

This paper introduces a novel kernel-based formulation of the Equalized Odds (EO) criterion, denoted as $\mathrm{EO}_k$, for fair representation learning (FRL) in supervised settings. The central goal of FRL is to mitigate discrimination regarding a sensitive attribute $S$ while preserving prediction accuracy for the target variable $Y$. Our proposed criterion enables a rigorous and interpretable quantification of three core fairness objectives: independence ($\widehat{Y} \perp\!\!\!\perp S$), separation–also known as equalized odds ($\widehat{Y} \perp\!\!\!\perp S \mid Y$), and calibration ($Y \perp\!\!\!\perp S \mid \widehat{Y}$). Under both unbiased ($Y \perp\!\!\!\perp S$) and biased ($Y \not\!\perp\!\!\!\perp S$) conditions, we show that $\mathrm{EO}_k$ satisfies both independence and separation in the former, and uniquely preserves predictive accuracy while lower bounding independence and calibration in the latter, thereby offering a unified analytical characterization of the tradeoffs among these fairness criteria. We further define the empirical counterpart, $\widehat{\mathrm{EO}}_k$, a kernel-based statistic that can be computed in quadratic time, with linear-time approximations also available. A concentration inequality for $\widehat{\mathrm{EO}}_k$ is derived, providing performance guarantees and error bounds, which serve as practical certificates of fairness compliance. While our focus is on theoretical development, the results lay essential groundwork for principled and provably fair algorithmic design in future empirical studies.

## 1 Introduction

As machine learning becomes increasingly integrated into social decision-making (e.g., hiring, lending), ensuring algorithmic fairness has emerged as a critical challenge. Consider an input random variable $X$ (e.g., applicant data), a protected attribute $S$ (e.g., gender, race), and a target variable $Y$ (e.g., loan default). A naïve approach might exclude $S$ from model training. However, when $X$ and $S$ are statistically dependent (e.g., due to historical biases in education or employment), models trained on $X$ alone can still perpetuate discrimination by leveraging proxy features correlated with $S$.

**Fair Representation Learning**. Fair Representation Learning (FRL) addresses this issue by constructing an intermediate representation $Z := f(X)$ that preserves task-relevant information for predicting the target attribute $Y$ while mitigating biases tied to the sensitive attribute $S$. Consider a binary classification task with target $Y \in \{0, 1\}$ and predictor $\widehat{Y} \sim \mathrm{Bernoulli}(h(Z))$. Existing approaches to evaluating discrimination in FRL primarily fall into three categories, each corresponding to a different statistical independence criterion:

39th Conference on Neural Information Processing Systems (NeurIPS 2025).

- **Independence** ($\widehat{Y} \perp\!\!\!\perp S$) seeks to eliminate the dependence of the protected attribute $S$ from the predictor, typically measured by Demographic Parity (DP, Dwork et al. [2012]), that is,

$$\mathrm{DP}(h; Z, Y, S) := |\Pr(\widehat{Y} = 1 \mid S = 1) - \Pr(\widehat{Y} = 1 \mid S = 0)|. \tag{1}$$

- **Separation** ($\widehat{Y} \perp\!\!\!\perp S \mid Y$) requires equal prediction performance conditioned on the true label $Y$, instantiated the by Equalized Odds (EO, Hardt et al. [2016]) constraint, i.e.,

$$\Pr(\widehat{Y} = 1 \mid Y = y, S = 0) = \Pr(\widehat{Y} = 1 \mid Y = y, S = 1), \quad y \in \{0, 1\}. \tag{2}$$

- **Calibration** ($Y \perp\!\!\!\perp S \mid \widehat{Y}$, Kleinberg [2018]) ensures the scoring function $h(Z)$ is equally meaningful across groups, demanding

$$\Pr(Y = 1 \mid h(Z) = t, S = 0) = \Pr(Y = 1 \mid h(Z) = t, S = 1),$$

which is equivalent to $Y \perp\!\!\!\perp S \mid \widehat{Y}$ (Barocas et al. [2023]). Proposed in Shen et al. [2022], a quantification of the calibration constraint, denoted as $\mathrm{DC}(h; Z, Y, S)$ for predictor $h$, is given as follows:

$$\frac{1}{4} \sum_{y \in \{0,1\}} \int_0^1 |\Pr(Y = y, h(Z) = t \mid S = 1) - \Pr(Y = y, h(Z) = t \mid S = 0)| dt. \tag{3}$$

**Conflicting Fairness Objectives**. Simultaneously satisfying independence, separation, and calibration is fundamentally infeasible under realistic conditions, even though each fairness constraint is individually well-justified. As shown in Kleinberg [2018], these criteria can only be satisfied together in pathological cases: either when the target variable $Y$ is independent of the sensitive attribute $S$, or when the predictor achieves perfect accuracy, i.e., $\Pr(\mathbb{E}[Y \mid Z] \in \{0, 1\}) = 1$. Such conditions are rarely encountered in practice, especially in FRL, where representations are often transferred to downstream tasks with unknown objectives. Consequently, practitioners are forced to navigate trade-offs between fairness definitions, often lacking clear guidance on the balance between competing objectives (Chouldechova [2016], Kleinberg [2018]). Identifying and characterizing these trade-offs is thus a critical step toward advancing the development of fair and effective representations.

**Accuracy Costs of Fairness**. Beyond their inherent incompatibility, fairness constraints can impose significant costs on predictive accuracy, presenting a fundamental trade-off in FRL. The objective in FRL is to construct representations that preserve task-relevant information for downstream prediction while suppressing dependencies on sensitive attributes. However, different fairness notions operationalize this objective in conflicting ways, often at the expense of accuracy. For instance, enforcing independence ($\widehat{Y} \perp\!\!\!\perp S$) removes all information correlated with the sensitive attribute $S$, including predictive features, thereby excluding the optimal classifier $\widehat{Y}^*$ that achieves perfect accuracy ($\Pr(\widehat{Y}^* = Y) = 1$, Hardt et al. [2016]) in cases where $Y \not\perp\!\!\!\perp S$. In contrast, the separation (EO) criteria address this limitation by conditioning fairness on the true label $Y$, enabling alignment with $\widehat{Y}^*$. Yet, such criteria may violate calibration, leading to systematic discrepancies in predicted probabilities across groups with the same outcome. These competing demands underscore the need for a principled and quantitative framework to assess how different fairness constraints interact and affect predictive performance — a gap this work aims to fill.

## 1.1 Main Contribution: A Quantification of Fairness–Accuracy Trade-offs

To systematically navigate the trade-offs between incompatible fairness constraints and their implications for predictive accuracy, we propose a kernel-based statistic, i.e., $\mathrm{EO}_k$, that quantifies these tensions within a unified framework. Our approach formalizes the divergence between conditional distributions associated with fairness notions such as independence, separation (EO), and calibration, allowing for precise measurement of how a given representation deviates from each criterion. Crucially, our statistic also reflects the extent to which fairness enforcement may distort task-relevant information, thereby linking fairness violations to potential accuracy degradation in downstream tasks. This dual role—diagnosing fairness violations while accounting for predictive utility—distinguishes our approach from existing methods. In what follows, we formally define the $\mathrm{EO}_k$ statistic, demonstrate its equivalence to a Maximum Mean Discrepancy (MMD) between reweighted group distributions, and analyze its theoretical properties—highlighting its expressiveness,

its ability to quantify fairness–accuracy trade-offs under dataset bias, and its practical computability. We also derive a generalization bound for its empirical counterpart and show how it can serve as a provable certificate for fairness-aware representation learning.

**Technical Overview of** $\mathrm{EO}_k$. Specifically, $\mathrm{EO}_k$ quantifies the maximum violation of the EO constraint $\widehat{Y} \perp\!\!\!\perp S \mid Y$ across predictors $h : \mathcal{Z} \mapsto [0, 1]$ derived from an affine map of the unit ball in a reproducing kernel Hilbert space (RKHS). Let $Z_s := Z \mid S = s$, and $p_{y|s} := \Pr(Y = y \mid S = s)$. Given a reproducing kernel $k : \mathcal{Z} \times \mathcal{Z} \to \mathbb{R}$ and its associated RKHS $\mathcal{H}_k$, the $\mathrm{EO}_k$ metric is defined as the supremum of class-weighted expectation differences within the unit ball of $\mathcal{H}_k$:

$$\mathrm{EO}_k(Z, Y, S) := \sup_{\|h\|_{\mathcal{H}_k} \leq 1} \left| p_{0|0} \mathbb{E}[h(Z_0 - Z_1) \mid Y = 0] + p_{1|0} \mathbb{E}[h(Z_0 - Z_1) \mid Y = 1] \right|, \tag{4}$$

where $\| \cdot \|_{\mathcal{H}_k}$ refers to the norm in RKHS $\mathcal{H}_k$ induced from kernel $k$. Let $Z_s^y := Z \mid S = s, Y = y$. The above definition is equivalent to the MMD considering random variables $\{p_{0|0} Z_s^0 + p_{1|0} Z_s^1\}_{s=0,1}$. As detailed in Lemma 2.2, an affine map of the unit ball in the RKHS formulates a feasible set, i.e., $\mathcal{H}$, of predictors $h : \mathcal{Z} \mapsto [0, 1]$ for the prediction of target variable $Y$. In other words, $\mathrm{EO}_k$ measures the worst-case deviation from the weighted EO constraint across all admissible predictors $h \in \mathcal{H}$.

**Expressiveness of Unit Ball in RKHS**. The affine-transformed RKHS unit ball can be considered as a feasible set for the downstream task that ensures expressiveness for capturing $Y \in \{0, 1\}$-dependent disparities, from the following two aspects:

- **Pratical Coverage**: By the representer theorem (Schölkopf et al. [2001]), an RKHS contains hypothesis classes of kernel Support Vector Machines (SVMs), Gaussian processes, and PCA, which are standard tools in the machine learning context.

- **Universal Approximation** With $c_0$-univeral kernels (e.g., Gaussian, Laplacian), the RKHS $\mathcal{H}_k$ densely spans the space of continuous functions vanishing at infinity (Sriperumbudur et al. [2010a]). That is, the elements in $\mathcal{H}_k$ can approximate any bounded continuous predictor, including neural networks, to arbitrary precision.

**Formalizing Fairness Trade-offs**. Unlike existing frameworks that require heuristic selection among mutually incompatible criteria—independence (1), EO (2), or calibration (3)—our context-adaptive $\mathrm{EO}_k$ automatically adjusts to the underlying dependency between $Y$ and $S$. Specifically, given a bounded reproducing kernel, i.e., $\sup_z k(z, z) \leq \nu$. For simplicity, let $\nu = 1/4$. In distinct $Y$-$S$ dependency structures, we have

- $Y \perp\!\!\!\perp S$: When no inherent bias exists ($Y$ independent of $S$), the minimization of $\mathrm{EO}_k$ enforces both the independence (1) and separation (EO) constraints (2). Formally, considering the feasible set, i.e., $\mathcal{H}$, of predictors for the downstream task, we have (Theorem 2.5)

$$\sup_{h \in \mathcal{H}} \mathrm{DP}(h; Z, Y, S) = \mathrm{EO}_k(Z, Y, S), \tag{5}$$

where $\mathcal{H}$ is derived from the affine image of the unit ball in an RKHS $\mathcal{H}_k$ (Lemma 2.2). To elaborate, (5) demonstrates that in bias-free regimes ($Y \perp\!\!\!\perp S$), the minimization of $\mathrm{EO}_k$ enforces both the EO (2) and the independence constraints (1) universally, $\forall h \in \mathcal{H}$. Here, $\mathcal{H}$ is the feasible set for predictors $h : \mathcal{Z} \mapsto [0, 1]$ implementable in downstream tasks.

- $Y \not\perp\!\!\!\perp S$: In the case of data bias ($Y$-$S$ dependency), $\mathrm{EO}_k$ permits a quantifiable accuracy-fairness trade-off: under mild constraints, there exists a constant $c$, such that (Theorem 2.5, 2.6)

$$c \sup_{h \in \mathcal{H}} \mathrm{DC}(h; Z, Y, S) \geq \sup_{h \in \mathcal{H}} \mathrm{DP}(h; Z, Y, S) \geq |p_{0|0} - p_{0|1}|\beta - \mathrm{EO}_k. \tag{6}$$

This inequality illustrates the incompatibility of fairness criteria: when a representation $Z$ nearly satisfies the EO constraint ($\mathrm{EO}_k \approx 0$) and retains nontrivial predictive power ($\beta > 0$), both independence (DP) and calibration (DC) constraints must necessarily be violated. Here, the coefficient $|p_{0|0} - p_{0|1}|$ is a quantification of the inherent dataset bias, and $\beta$ represents the optimal balanced accuracy achievable under $S = 1$ (Lemma 2.4):

$$\sup_{h \in \mathcal{H}} \frac{1}{2}(\Pr(\widehat{Y} = 0 \mid Y = 0, S = 1) + \Pr(\widehat{Y} = 1 \mid Y = 1, S = 1)) \leq \frac{1 + \beta}{2}.$$

A value of $\beta > 0$ implies that $Z$ supports better-than-random prediction. Moreover, enforcing the calibration constraint (3) for all downstream predictors $h \in \mathcal{H}$ entails satisfying independence, thus

imposing a stricter condition than independence alone. This makes calibration particularly costly in terms of preserving task-relevant signal. Lastly, since the Bayes-optimal classifier $\widehat{Y}^*$—which achieves perfect accuracy, i.e., $\Pr(\widehat{Y}^* = Y) = 1$—satisfies the EO constraint but violates the DP constraint (1) (Hardt et al. [2016]), the DP (and hence calibration) constraints are fundamentally at odds with predictive accuracy in biased settings. In contrast, EO, as quantified by our $\mathrm{EO}_k$ statistic, accommodates nontrivial predictive performance while explicitly characterizing fairness–accuracy trade-offs, making it the most appropriate constraint in such contexts.

**Scalable Estimation via MMD**. Our $\mathrm{EO}_k$ statistic admits a closed-form empirical estimator based on the MMD framework, enabling efficient evaluation. More specifically, let $\bar{Z}^{(s)} := p_{0|0} Z_s^0 + p_{1|0} Z_s^1$ denote the $Y$-reweighted mixture distribution for group $S = s$, where $s \in \{0, 1\}$. Given $n_0$ and $n_1$ i.i.d. samples $\{z_i^{(0)}\}_{i=1}^{n_0}$ and $\{z_i^{(1)}\}_{i=1}^{n_1}$ from $\bar{Z}^{(0)}$ and $\bar{Z}^{(1)}$ respectively, the empirical estimator for $\mathrm{EO}_k$ is given as follows:

$$\widehat{\mathrm{EO}}_k := \sqrt{\frac{\sum_{i \neq j}^{m} k(z_i^{(0)}, z_j^{(0)})}{n_0(n_0 - 1)} + \frac{\sum_{i \neq j}^{n} k(z_i^{(1)}, z_j^{(0)})}{n_1(n_1 - 1)} - \frac{2 \sum_{i,j=1}^{n_0, n_1} k(z_i^{(0)}, z_j^{(1)})}{n_0 n_1}}. \tag{7}$$

This formulation enables direct integration with Stochastic Gradient Descent (SGD, Briol et al. [2019], Rychener et al. [2022]) and is evaluable in $O(n^2)$ (Gretton et al. [2012]) or even $O(n)$ time (Zhao and Meng [2015]), where $n := n_0 + n_1$ refers to the sample size of for the pairs $(X, S, Y)$. In practice, one can implement this i.i.d. sampling requirement via stratified bootstrap resampling.

**Generalization Guarantees and Domain Adaptation**. Our empirical estimator $\widehat{\mathrm{EO}}_k$ is equipped with hyperparameter-free convergence when served as a penalty term of the objective function in FRL, ensuring that the employment of $\widehat{\mathrm{EO}}_k$ can help us justify the achievement for $\mathrm{EO}_k$. Specifically, building upon the uniform MMD error bound from Ni and Huo [2024], we prove a non-asymptotic error bound that holds uniformly over penalty coefficients and optimization trajectories, guaranteeing convergence of the empirical estimator to the population-level $\mathrm{EO}_k$ constraint as sample size increases. As a representative example, suppose the encoders $f \in \mathcal{F}$ mapping from the input $X$ to the representation $Z$ are composed of feed-forward neural networks, $\forall \delta \in (0, 1)$, we have (Theorem 2.10)

$$\Pr\left(\sup_{f \in \mathcal{F}} \left|\widehat{\mathrm{EO}}_k^2 - \mathrm{EO}_k^2\right| \leq O\left(\sqrt{\frac{\log(d) + \log(\delta^{-1})}{n_0 + n_1}}\right)\right) \geq 1 - \delta,$$

where $n_0, n_1$ refers to the sample sizes for $\bar{Z}^{(0)}$ and $\bar{Z}^{(1)}$, respectively, $d$ refers to the input dimension. The logarithmic relationship between the deviation bound and the input dimension underscores EO's computational edge over traditional Integral Probability Metrics (IPMs) in high-dimensional tasks. Moreover, in cases where the input $X$ is perturbed via a function transformation $g(X)$, suppose $f \circ g \in \mathcal{F}, \forall \mathcal{F}$, the above upper bound remains valid, revealing the domain adaptation property of the proposed metric $\mathrm{EO}_k$.

## 1.2 Related Works: The Limits of Achieving Multiple Group Fairness

To address the ambiguity in selecting a fairness criterion, recent works have proposed metrics that aim to approximate multiple fairness notions simultaneously. For example, it is proposed in Shen et al. [2022] that the minimization through the following opposing objectives leads to the joint approximation of group fairness constraints embedded in independence (1), separation (2), and calibration (3). That is,

$$\max\{d_{\mathrm{TV}}(Z_0, Z_1), 1 - d_{\mathrm{TV}}(Z^0, Z^1)\},$$

where $Z_s := Z \mid S, Z^y := Z \mid Y = y$, $d_{\mathrm{TV}}$ refers to the Total Variation Distance (TVD). Similarly, Jang et al. [2024] design a metric lower bounded by both DP (1) and EO (2) violations.

However, these approaches conflict with fundamental impossibility results (Kleinberg [2018]), which show that independence, separation, and calibration cannot be satisfied simultaneously except in degenerate cases (e.g., $Y \perp\!\!\!\perp S$ or perfect prediction). In particular, the formulation in Shen et al. [2022] assumes that the involved conflicting TVD terms, i.e., $d_{\mathrm{TV}}(Z_0, Z_1)$ and $d_{\mathrm{TV}}(Z^0, Z^1)\}$, can be simultaneously optimized, despite their inherent trade-off under distributional bias ($Y \not\!\perp\!\!\!\perp S$), where the composite metric can never be minimized to zero. Similarly, the metric proposed in Jang

et al. [2024] fails to account for the structural incompatibility between the DP (1) and EO (2) metric in biased settings ($Y \not\perp\!\!\!\perp S$). As shown in (6), both criteria can only be simultaneously satisfied when the target variable becomes unidentifiable in subgroup $S = 1$. Consequently, to preserve the prediction accuracy, the lower bound of the proposed metric is determined by the degree of DP, which remains strictly positive under $Y \not\perp\!\!\!\perp S$.

In contrast, our method acknowledges this conflict and formalizes the trade-offs using a kernel-based statistic, $\mathrm{EO}_k$. It can be observed that our metric $\mathrm{EO}_k$ preserves the Bayes-optimal predictor in biased regimes and approximates both independence and EO in the unbiased case, providing a principled approach to fairness constraint selection based on task-specific trade-offs.

## 1.3 Preference of MMD in FRL

**Adversarial Training Structure in FRL**. Adversarial training is an approach widely used in FRL (Beutel et al. [2017], Madras et al. [2018], Zhao et al. [2019], Kim and Cho [2020]) considering the accuracy and fairness trade-off. Let an encoder $f : \mathcal{X} \mapsto \mathcal{Z}$ map features $X$ to a representation $Z = f(X)$. A task head $h : \mathcal{Z} \mapsto [0, 1]$ predicts the target attribute $\widehat{Y} \sim \mathrm{Bernoulli}(h(Z))$, while a discriminator $d : \mathcal{Z} \mapsto [0, 1]$ tries to recover the protected attribute $S$ from $Z$. The resulting min-max problem is

$$\min_{f,h} \max_d \mathbb{E}\big[\mathcal{L}_Y(h(Z), Y)\big] - \lambda \, \mathbb{E}\big[\mathcal{L}_S(d(Z), S)\big], \tag{8}$$

where $\mathcal{L}_Y$ (e.g., cross-entropy) encourages predictive accuracy, $\mathcal{L}_S$ (e.g., logistic loss) penalizes information about $S$ in $Z$, and $\lambda > 0$ controls the trade-off. At equilibrium, $Z$ is approximately independent of $S$, thereby approaching DP, i.e., $h(Z) \perp\!\!\!\perp S$, for any downstream prediction model $h$. For a comprehensive survey of FRL approaches, readers are referred to Cerrato et al. [2024].

**Reducing Computation via IPM penalty**. Replacing the adversarial maximization step in (8) with an Integral Probability Metric (IPM) regularizer has gained wide attention in recent studies (Mary et al. [2019], Kim et al. [2022], Deka and Sutherland [2023], Kong et al. [2025]). Examples include TVD (Shen et al. [2022]), MMD (Oneto et al. [2020], Rychener et al. [2022], Deka and Sutherland [2023]), and the Wasserstein Distance (Gordaliza et al. [2019]). Notably, an IPM is defined as a supremum over a specified function class $\mathcal{F}$ of real-valued functions. Specifically, given a pair of random variables $Z_0$ and $Z_1$ embedded in set $\mathcal{Z}$, we have

$$d_{\mathcal{F}}(Z_0, Z_1) := \sup_{f \in \mathcal{F}} \left| \mathbb{E}[f(Z_0)] - \mathbb{E}[f(Z_1)] \right|,$$

which collapses the inner optimization into a single, closed-form loss term, eliminating the full min-max game and reducing computational overhead. This substitution also yields tighter theoretical guarantees for DP (Kong et al. [2025]).

**From TVD to MMD: A Practical Shift in Fairness Regularization**. For binary classification, i.e., $Y \in \{0, 1\}$, TVD is the theoretically ideal IPM for enforcing demographic parity. Its specified function class $\mathcal{F}_{\mathrm{TV}} := \{f : \mathcal{Z} \to [-1, 1]\}$ is an affine transformation of the set $\mathcal{H}_{[0,1]} := \{h : \mathcal{Z} \to [0, 1]\}$ that comprises all probabilistic classifiers for the event $Y = 1$. Despite its theoretical appeal, TVD suffers from critical limitations in practice. Most notably, its gradients vanish when the supports of the conditional distributions do not overlap—stalling optimization in high-dimensional or imbalanced settings. This makes TVD ill-suited for gradient-based training in deep representation learning frameworks. These limitations have led to the widespread adoption of MMD as a computationally efficient and stable alternative.

# 2 Main Theoretical Results

In the following, we first provide the formal definition of our metric $\mathrm{EO}_k$ and its technical explanation in Section 2.1. Given the formal definition, we discuss the relationships between our metric and the independence and calibration constraints in Section 2.2, showing that in both the biased ($Y \perp\!\!\!\perp S$) and unbiased ($Y \not\perp\!\!\!\perp S$) cases, our metric is a preferred choice. Regarding the algorithmic implementation, we provide the empirical estimator of our metric and discuss its convergence rate when served as a penalty term in the objective function in Section 2.3.

## 2.1 Kernel-based EO Constraint

Given a characteristic reproducing kernel (Definition 4.2), from which the derived kernel mean embeddings are injective, the EO constraint ($\widehat{Y} \perp\!\!\!\perp S \mid Y$, (2)) in the binary classification setting is equivalent to the following condition:

$$\mu_0^y = \mu_1^y, \quad y \in \{0, 1\},$$

where $\mu_s^y := \mathbb{E}[k(\cdot, Z_s^y)]$ stands for the kernel mean embedding of the conditional representation $Z_s^y$, $s, y \in \{0, 1\}$. In the following, instead of computing the MMD for both the $(\mu_0^0, \mu_1^0)$ and $(\mu_0^1, \mu_1^1)$ pairs, we consider the difference between the following weighted summation conditioned on $Y = 0$ and $Y = 1$, i.e.,

$$p_{0|0}\mu_s^0 + p_{1|0}\mu_s^1,$$

for $s = 0, 1$, where $p_{y|s} := \Pr(Y = y | S = s)$, $s, y \in \{0, 1\}$. The formal definition is given as follows:

**Definition 2.1** (Weighted Equalized Odds via MMD). *Let $\mathcal{H}_k$ be an RKHS containing functions mapping from $\mathcal{Z}$ to $\mathbb{R}$, and $k : \mathcal{Z} \times \mathcal{Z} \mapsto \mathbb{R}$ be the corresponding reproducing kernel. Suppose $k$ is characteristic as defined in Definition 4.3, we quantify the separation constraint, i.e., $\widehat{Y} \perp\!\!\!\perp S \mid Y$, through the following expression:*

$$\mathrm{EO}_k(Z, Y, S) := \gamma_k(p_{0|0}Z_0^0 + p_{1|0}Z_0^1, p_{0|0}Z_1^0 + p_{1|0}Z_1^1). \tag{9}$$

## 2.2 Accuracy-Fairness Trade-off

We start with the statement that the value of MMD can be considered as the supremum of the balanced accuracy for a binary classification problem. In the following, we provide the involved function class, built from an affine map of the unit ball in an RKHS.

**Lemma 2.2.** *Let $\mathcal{H}_k$ be an RKHS containing functions mapping from $\mathcal{Z}$ to $\mathbb{R}$, and $k : \mathcal{Z} \times \mathcal{Z} \mapsto \mathbb{R}$ be the corresponding reproducing kernel. Suppose $\sup_z k(z, z) \leq \nu$. Let $\mathcal{H}_{[0,1]} := \{h : \mathcal{Z} \mapsto [0, 1]\}$ be the set containing all possible classifiers, $\mathcal{H} := \{h \mid h(z) = (h_k(z) + 1)/2, \|h_k\|_{\mathcal{H}_k} \leq \nu^{-1/2}\}$, we have $\mathcal{H} \subseteq \mathcal{H}_{[0,1]}$.*

Proof of the above lemma is given in Appendix 4.2.

The formal definition of the balanced accuracy for a binary classification problem is given as follows:

**Definition 2.3** (Balanced Accuracy). *Given an input random variable $X \in \mathcal{X}$ and a binary target attribute $Y \in \{0, 1\}$. Let $\widehat{Y} \sim \mathrm{Bernoulli}(h(X))$ be the prediction derived based on $X$, where $h : \mathcal{X} \mapsto [0, 1]$. The Balanced Accuracy (BA) of the predictor $h$ is measured by the following equation:*

$$\mathrm{BA}(h; X, Y) := \frac{1}{2}\left(\Pr(\widehat{Y} = 0 \mid Y = 0) + \Pr(\widehat{Y} = 1 \mid Y = 1)\right).$$

Considering the aforementioned function set, the following lemma provides the relationship between the value of MMD and the optimal balanced accuracy.

**Lemma 2.4** (MMD and Optimal Balanced Accuracy). *Let $\mathcal{H}_k$ be an RKHS containing functions mapping from $\mathcal{Z}$ to $\mathbb{R}$, and $k : \mathcal{Z} \times \mathcal{Z} \mapsto \mathbb{R}$ be the corresponding reproducing kernel. Suppose $\sup_z k(z, z) \leq \nu$, let $\mathcal{H} := \{h \mid h(z) = (h_k(z) + 1)/2, \|h_k\|_{\mathcal{H}_k} \leq \nu^{-1/2}\}$. Given a representation $Z$ embedded in $\mathcal{Z}$, a sensitive attribute $S \in \{0, 1\}$, and a target attribute $Y \in \{0, 1\}$. Then, the MMD between the conditional representation $Z_s^y$ corresponds to the optimal balanced accuracy with respect to the classifiers embedded in $\mathcal{G}$. More specifically, we have*

*1. Suppose $\gamma_k(Z_0, Z_1) \leq \alpha$, then*

$$\sup_{h \in \mathcal{H}} \mathrm{BA}(h; Z, S) \leq \frac{2 + \nu^{-1/2}\alpha}{4}.$$

*2. Suppose $\gamma_k(Z^0, Z^1) \geq \beta$, then*

$$\sup_{h : \mathcal{Z} \mapsto [0,1]} \mathrm{BA}(h; Z, Y) \geq \frac{2 + \nu^{-1/2}\beta}{4}.$$

*Here, $\alpha \geq 0$ and $\beta \leq 2\nu^{1/2}$.*

Proof of the above lemma is referred to Appendix 4.3.

Observed from the above lemma, it can be concluded that (i) The minimization of MMD over representation $Z$ conditioned on the sensitive attribute $S$ enforces the statistical indistinguishability of predictions $\widehat{Y} \sim \text{Bernoulli}(h(Z))$ across sensitive groups, as quantified by the maximal predictive leakage of $S$ under optimal balanced accuracy; (ii) To certify non-trivial predictive performance in downstream tasks, the MMD over representations $Z$ conditioned on the target attribute $Y$ must be bounded below by a positive constant. This guarantees the existence of a predictor $h : \mathcal{Z} \mapsto [0, 1]$ whose induced prediction $\widehat{Y} \sim \text{Bernoulli}(h(Z))$ can surpass random guessing.

In the following, we analyze the relationship between the independence constraint (1) and the metric $\text{EO}_k$ that represents the EO constraint (2), considering the supremum over the possible predictors $h \in \mathcal{H}$, where $\mathcal{H}$ is the feasible set induced by the unit ball in an RKHS as given in Lemma 2.2. Combining the following result and the above observation regarding the relationship between MMD and prediction accuracy, we conclude that the metric $\text{EO}_k$ is a preferred choice considering the fairness-accuracy trade-off.

**Theorem 2.5** (Independence and EO). *Let $\mathcal{H}_k$ be an RKHS containing functions mapping from $\mathcal{Z}$ to $\mathbb{R}$, and $k : \mathcal{Z} \times \mathcal{Z} \mapsto \mathbb{R}$ be the corresponding reproducing kernel. Suppose $\sup_z k(z, z) \leq \nu$. Let $\mathcal{H} := \{h \mid h(z) = (h_k(z) + 1)/2, \|h_k\|_{\mathcal{H}_k} \leq \nu^{-1/2}\}$, DP be given as (1), we have*

$$\sup_{h \in \mathcal{H}} \text{DP}(h; Z, Y, S) = (2\nu^{1/2})^{-1} \gamma_k(Z_0, Z_1). \tag{10}$$

*Moreover, denote $\Pr(Y = y \mid S = s)$ as $p_{y|s}$. Let $\text{EO}_k$ be the weighted equalized odds constraint given in Definition 2.1. Suppose $Y \perp\!\!\!\perp S$, we have*

$$\sup_{h \in \mathcal{H}} \text{DP}(h; Z, Y, S) = (2\nu^{1/2})^{-1} \text{EO}_k(Z, Y, S). \tag{11}$$

*In cases where $Y \not\perp\!\!\!\perp S$, we have*

$$\sup_{h \in \mathcal{H}} \text{DP}(h; Z, Y, S) \geq (2\nu^{1/2})^{-1} \left| |p_{0|0} - p_{0|1}| \gamma_k(Z_1^0, Z_1^1) - \text{EO}_k(Z, Y, S) \right|. \tag{12}$$

Proof of the above theorem is referred to Appendix 4.4.

This result underscores the inherent incompatibility between independence (DP) and separation (EO) constraints when $Y \not\perp\!\!\!\perp S$. Formally, if an optimization procedure yields $\text{EO}_k \approx 0$, the following lower bound holds:

$$\sup_{h \in \mathcal{H}} \text{DP} \geq (2\nu)^{-1} |p_{0|0} - p_{0|1}| \beta,$$

where $\beta := \gamma_k(Z_1^0, Z_1^1)$. Based on Lemma 2.4, $\gamma_k(Z_1^0, Z_1^1)$ quantifies the maximum achievable balanced accuracy for predicting $Y$ given $S = 1$. To ensure non-trivial predictive performance, i.e., the balanced accuracy is greater than $1/2$, we require $\beta > 0$, which strictly prohibits perfect independence, i.e., $\text{DP} = 0$ in the biased setting ($Y \not\perp\!\!\!\perp S$), aligning with the impossibility result proposed in Hardt et al. [2016], Kleinberg [2018]. In conclusion, we have (i) the minimization of the metric $\text{EO}_k$ achieves both the independence and EO constraint simultaneously in the unbiased setting where $Y \perp\!\!\!\perp S$; (ii) the minimization of $\text{EO}_k$ is preferred in the biased setting $Y \not\perp\!\!\!\perp S$ to ensure the prediction accuracy regarding the target attribute $Y$.

**Theorem 2.6** (Independence and Calibration). *Given characteristic reproducing kernels $k : \mathbb{R}^{|Z|} \times \mathbb{R}^{|Z|} \mapsto \mathbb{R}$, $k_{[0,1]} : [0, 1] \times [0, 1] \mapsto \mathbb{R}$, and $k_{\{0,1\}} : \{0, 1\} \times \{0, 1\} \mapsto \mathbb{R}$. Let $\mathcal{H}$, $\mathcal{H}_{[0,1]}$, and $\mathcal{H}_{\{0,1\}}$ be the corresponding RKHSs. Let $\mathcal{H}_{[0,1]} \otimes \mathcal{H}_{\{0,1\}}$ be the tensor product of space $\mathcal{H}_{[0,1]}$ and $\mathcal{H}_{\{0,1\}}$, equipped with a reproducing kernel function $k_{[0,1]} \otimes k_{\{0,1\}}$, where $k_{[0,1]} \otimes k_{\{0,1\}}((u, y), (u, y')) = k_{[0,1]}(u, u')k_{\{0,1\}}(y, y'), \forall u, u' \in [0, 1], \forall y, y' \in \{0, 1\}$.*

*Suppose $\sup_z k(z, z) \leq \nu$, $\sup_u k_{[0,1]}(u, u) \leq \nu_{[0,1]}$, $\sup_y k_{\{0,1\}}(y, y) \leq \nu_{\{0,1\}}$, and $\text{id} \in \mathcal{H}_{[0,1]}$, where $\text{id}$ is the identity function. Let $\mathcal{H} := \{h \mid h(z) = (h_k(z) + 1)/2, \|h_k\|_{\mathcal{H}_k} \leq \nu^{-1/2}\}$. We have*

$$\begin{aligned}
&\sup_{h \in \mathcal{G}} \text{DC}(h, Y, S) \\
\geq\ & (4\nu_{[0,1]}^{1/2} \nu_{\{0,1\}}^{1/2})^{-1} \sup_{h \in \mathcal{G}} \gamma_{k_{[0,1]} \otimes k_{\{0,1\}}}((Y, h(Z)) \mid S = 0, (Y, h(Z)) \mid S = 1) \\
\geq\ & (4\nu_{[0,1]}^{1/2} \nu_{\{0,1\}}^{1/2} \|\text{id}\|_{\mathcal{H}_{[0,1]}})^{-1} \sup_{h \in \mathcal{H}} \text{DP}(h; Z, Y, S).
\end{aligned}$$

Proof of the above theorem is referred to Appendix 4.5.

As observed from the above theorem, the calibration constraint (DC) is linearly lower bounded by the independence (DP) constraint. Consequently, in biased regimes ($Y \not\perp S$), achieving calibration requires suppressing DP below a problem-dependent threshold. This creates a two-fold conflict:

1. **Accuracy-Fairness Paradox**: An optimal predictor $\widehat{Y}^*$, such that $\Pr(\widehat{Y}^* = Y) = 1$, violates the independence constraint, i.e., DP $> 0$, leading to the violation for DC constraint.

2. **Violation of the EO constraint**: Recall the contradiction between the EO and independence constraint as given in Theorem 2.5. In a biased circumstance ($Y \not\perp S$), requiring DC $\approx 0$ leads to a lower bound for the EO constraint as measured by $\text{EO}_k$, corresponding to the fact that given the target attribute $Y$, the prediction made by the model relies on the sensitive attribute $S$.

Thus, in the biased setting $Y \not\perp S$, $\text{EO}_k$ emerges as the preferred fairness criterion over both DP and DC.

## 2.3 Optimization Framework with Convergence Guarantees

Utilizing the aforementioned metric $\text{EO}_k$ as a regularization constraint, we formulate the supervised FRL problem as follows:

$$\arg\min_{h,f} \mathcal{L}_{\text{sup}}(h \circ f) + \lambda \widehat{\text{EO}}_k^2, \quad \text{s.t. } Z = f(X), \tag{13}$$

where $\mathcal{L}_{\text{sup}}$ is a given supervised risk such as the cross-entropy in this binary classification setting, $\widehat{\text{EO}}_k$ refers to the empirical estimate of $\text{EO}_k$ as given in (7), and the multiplier $\lambda \geq 0$ is a hyperparameter controlling the relative magnitude of the fairness constraint. Here, to circumvent gradient instability caused by the square root function contained in the expression of $\widehat{\text{EO}}_k$, we employ $\widehat{\text{EO}}_k^2$ in the constraint instead.

To validate the empirical estimator $\widehat{\text{EO}}_k$'s efficacy as a penalty term, we provide a uniform deviation bound between $\widehat{\text{EO}}_k$ and $\text{EO}_k$ under the following assumptions. First, regarding the encoder set $\mathcal{F}$ containing mappings from $\mathcal{X}$ to $\mathcal{Z}$ to establish a representation $Z$, we require that its covering number is finite.

**Assumption 2.7.** *Let $\mathcal{F}$ be a function set mapping from $\mathcal{X}$ to $\mathcal{Z}$, where $\mathcal{Z} \subseteq \mathbb{R}^d$. $\forall \epsilon > 0$, suppose the covering number $N(\epsilon; \mathcal{F}, \| \cdot \|_\infty) < \infty$, where $\| \cdot \|_\infty$ is a function norm defined as follows:*

$$\|f\|_\infty := \inf \left\{ C \geq 0 \,\middle|\, \sup_{x \in \mathcal{X}} |f(x)| \leq C \right\}.$$

*The covering number $N(\epsilon; \mathcal{F}, \| \cdot \|)$ is defined as the minimum cardinality of a set $\mathcal{F}_\epsilon$, such that $\forall f \in \mathcal{F}$, there exists a function $f_\epsilon \in \mathcal{F}_\epsilon$, such that $\|f_\epsilon - f\| \leq \epsilon$.*

Second, regarding the reproducing kernel involved in the establishment of the empirical estimator, we assume it is bounded and Lipschitz.

**Assumption 2.8.** *Let $\mathcal{H}$ be an RKHS containing functions mapping from $\mathcal{Z}$ to $\mathbb{R}$, and $k : \mathcal{Z} \times \mathcal{Z} \mapsto \mathbb{R}$ be the corresponding reproducing kernel. Suppose the reproducing kernel $k$ is built under the following regularity conditions:*

*1. (**Bounded**). $\exists \nu > 0$, s.t., $\sup_{z \in \mathcal{Z}} k(z, z) \leq \nu$.*

*2. (**Lipschitz**). $\exists l > 0$, s.t., $\forall z \in \mathcal{Z}$, the function $k(\cdot, z) \in \mathcal{H}$ is $l$-Lipschitz.*

In the following, we provide the definition of Gaussian complexity, which will be a quantification considered in the deviation bound.

**Definition 2.9** (Gaussian Complexity). *The Gaussian complexity of a set $\mathcal{T} \subseteq \mathbb{R}^n$ is defined as follows:*

$$G_n(\mathcal{T}) := \mathbb{E}_\xi \sup_{\mathbf{t} \in \mathcal{T}} \left[ \sum_{i=1}^{n} \xi_i t_i \right], \tag{14}$$

*where $\mathbf{t} = (t_1, \ldots, t_n)^T$, $\xi_i \overset{i.i.d.}{\sim} \mathcal{N}(0, 1)$, $\forall i$.*

Based on Assumption 2.7 and 2.8, the technical statement for the uniform deviation bound is detailed as follows:

**Theorem 2.10** (Uniform Concentration Inequality of MMD, Ni and Huo [2024]). *Given an encoder set $\mathcal{F}$ containing mappings from $\mathcal{X}$ to $\mathcal{Z}$, where $\mathcal{Z} \subseteq \mathbb{R}^d$ and $Z = f(X)$ given a specified encoder $f \in \mathcal{F}$. Let $\overline{f(X)}^{(s)} := p_{0|0} f(X)_s^0 + p_{1|0} f(X)_s^1$ denote the $Y$-reweighted mixture distribution for group $S = s$, where $s \in \{0, 1\}$. Given $n_0$ and $n_1$ i.i.d. samples $\overline{f(\mathbf{X})}^{(0)} := \{\overline{f(x_i)}^{(0)}\}_{i=1}^{n_0}$ and $\overline{f(\mathbf{X})}^{(1)} := \{\overline{f(x_i)}^{(1)}\}_{i=1}^{n_1}$ from $\overline{f(X)}^{(0)}$ and $\overline{f(X)}^{(1)}$, respectively. Denote $n := n_0 + n_1$ ,$\rho_0 := n_0/n$, and $\rho_1 := n_1/n$. Given a reproducing kernel $k : \mathcal{Z} \times \mathcal{Z} \mapsto \mathbb{R}$, where $\mathcal{Z} \in \mathbb{R}^d$. Let $G_{nd}(\mathcal{F}(\overline{\mathbf{X}}))$ be the empirical Gaussian complexity defined in (14) regarding the set $\mathcal{F}(\overline{\mathbf{X}}) := \{(\overline{f(x_1)}^{(0)}, \ldots, \overline{f(x_{n_0})}^{(0)}, \overline{f(x_0)}^{(1)}, \ldots, \overline{f(x_{n_1})}^{(0)}\}$. $\forall \delta \in (0, 1)$, with probability at least $1 - \delta$, we have*

$$
\sup_{f \in \mathcal{F}} \left| \widehat{\mathrm{EO}}_k^2(\overline{f(\mathbf{X})}) - \mathrm{EO}_k^2(\overline{f(X)}) \right|
$$

$$
\leq 8\nu \max\left\{\rho_0^{-1}, \rho_1^{-1}\right\} \sqrt{\frac{\log(2/\delta)}{n}} + \frac{2\sqrt{2\pi}l}{n} \max\left\{\frac{1 + \rho_0^{-1}}{\rho_0}, \frac{1 + \rho_1^{-1}}{\rho_1}\right\} \mathbb{E}\left[G_{nd}(\mathcal{F}(\overline{\mathbf{X}}))\right];
$$

Notably, the error bound established in the above theorem relies on the Gaussian complexity of the involved encoder set, whose computation involved a supremum with respect to $f \in \mathcal{F}$. In the following, we provide an error bound for the empirical estimator we obtained in (13). As a representative example, we establish the deviation bounds for neural network scenario as discussed in Proposition 4.7.

**Corollary 2.11** (Error Bounds for Empirical MMD). *Let $\mathcal{F}$ be a function set mapping from $\mathcal{X}$ to $\mathcal{Z}$, where $\mathcal{Z} \subseteq \mathbb{R}^d$. Let $f_n^*$ be the empirical estimator obtained in the optimization problem (13), corresponding to the representation $Z = f_n^*(X)$ and its data matrices $\overline{f_n^*(\mathbf{X})}^{(0)}$ and $\overline{f_n^*(\mathbf{X})}^{(0)}$, as defined in Theorem 2.10. Under the Assumption 2.8, there exist constants $C_1$, $C_2$, such that $\forall \delta \in (0, 1)$, with probability at least $1 - \delta$, we have*

$$
\mathrm{EO}_k^2(\overline{f(X)}) \leq \widehat{\mathrm{EO}}_k^2(\overline{f(\mathbf{X})}) + \frac{C_1}{n} \mathbb{E}\left[G_{nd}(\mathcal{F}(\overline{\mathbf{X}}))\right] + C_2 \sqrt{\frac{\ln(2/\delta)}{n}}.
$$

*Suppose $\mathcal{F}$ is a set of feed-forward neural networks as given in Proposition 4.7. $\forall \delta \in (0, 1)$, with probability at least $1 - \delta$, we have*

$$
\mathrm{EO}_k^2(\overline{f(X)}) \leq \widehat{\mathrm{EO}}_k^2(\overline{f(\mathbf{X})}) + \mathcal{O}\left(n^{-1/2}\left(1 + \sqrt{\ln(2/\delta)}\right)\right).
$$

Based on the above result, suppose $\widehat{\mathrm{EO}}_k^2(\overline{f(\mathbf{X})}) \leq \epsilon$ in the feed-forward neural network scenario, then $\forall \delta \in (0, 1)$, with probability at least $1 - \delta$, its expected value, i.e., the $\mathrm{EO}_k$ metric, can be controlled by is $\sqrt{\epsilon^2 + \mathcal{O}\left(n^{-1/2}\left(1 + \sqrt{\ln(2/\delta)}\right)\right)}$, confirming the convergence rate and the efficacy of the empirical estimator $\widehat{\mathrm{EO}}_k$ when adopted as a penalty term in FRL.

## 3 Conclusion

We present a kernel-based statistic, $\mathrm{EO}_k$, that quantifies the trade-offs between incompatible fairness constraints and predictive accuracy in representation learning. Unlike prior work that attempts to approximate multiple fairness notions simultaneously, our approach acknowledges their inherent conflict and provides a principled way to measure and navigate it. Our metric $\mathrm{EO}_k$ admits a scalable empirical estimator, recovers DP (Independence, Equation (1)) and EO (Separation, Equation (2)) simultaneously in unbiased settings, and serves as the one that preserves the Bayes-optimal predictor under bias setting when compared to DP and DC (Calibration, Equation (3)) metrics. This framework offers a practical and theoretically grounded tool for fairness constraint selection in real-world applications. Our current work is theoretical. However, it lays the foundation for future empirical studies.

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

# 4 Appendix

In the appendix, we will first provide the preliminaries to prepare for the proofs we will discuss. More specifically, we first provide Table 1 to give an overview of formal definitions of existing quantifications for fairness constraints. Then, we provide the formal definition of MMD and RKHS, and discuss the relationship between MMD and TVD.

Afterwards, we provide the proofs for Lemma 2.2, 2.4, Theorem 2.5, 2.6, and 4.5.

Finally, we present the concentration inequality of Gaussian complexity and its upper bound for neural networks as supplementary material for our established error bounds.

To begin with, Table 1 provides a quantification of the aforementioned fairness constraints in a binary classification setting, i.e., $Y \in \{0, 1\}$.

## 4.1 Preliminary: RKHS and MMD

We review the basic idea behind MMD, which is to quantify the discrepancy between the two distributions $P$ and $Q$ in terms of the largest difference in expectation between $f(X)$ and $f(Y)$, for $X \sim P$ and $Y \sim Q$, over functions $f$ in the unit ball of a reproducing kernel Hilbert space (RKHS) defined on $\mathcal{X}$. This is called the maximum mean discrepancy (MMD) between distributions $P$ and $Q$, which can be conveniently estimated from the data in terms of the pairwise kernel dissimilarities. For characteristic kernels, a useful property of the MMD is that it takes value zero if and only if

Table 1: Existing fairness notions in a binary classification task [Shen et al., 2022, Table 1]. $\widehat{Y} \sim \text{Bernoulli}(h(Z))$, $Y$, and $S$ stand for empirical predictor, true label, and sensitive attribute, respectively.

| Category | Definition | |
|---|---|---|
| $\widehat{Y} \perp\!\!\!\perp S$ | $\text{DP}(\widehat{Y}, S) =$ | $\|\Pr(\widehat{Y} = 1 \mid S = 1) - \Pr(\widehat{Y} = 1 \mid S = 0)\|$ |
| $\widehat{Y} \perp\!\!\!\perp S \mid Y$ | $\text{DOpp}(\widehat{Y}, Y, S) =$ | $\|\Pr(\widehat{Y} = 1 \mid Y = 1, S = 1) - \Pr(\widehat{Y} = 1 \mid Y = 1, S = 0)\|$ |
| | $\text{DR}(\widehat{Y}, Y, S) =$ | $\|\Pr(\widehat{Y} = 1 \mid Y = 0, S = 1) - \Pr(\widehat{Y} = 1 \mid Y = 0, S = 0)\|$ |
| | $\text{DOdds}(\widehat{Y}, Y, S) =$ | $\frac{1}{2} \times (\text{DOpp}(\widehat{Y}, Y, S) + \text{DR}(\widehat{Y}, Y, S))$ |
| $Y \perp\!\!\!\perp S \mid \widehat{Y}$ | $\text{DPC}(h, Y, S) =$ | $\frac{1}{2} \sum_{t \in [0,1]} \|\Pr(Y = 1, h(Z) = t \mid S = 1) - \Pr(Y = 1, h(Z) = t \mid S = 0)\|$ |
| | $\text{DNC}(h, Y, S) =$ | $\frac{1}{2} \sum_{t \in [0,1]} \|\Pr(Y = 0, h(Z) = t \mid S = 1) - \Pr(Y = 0, h(Z) = t \mid S = 0)\|$ |
| | $\text{DC}(h, Y, S) =$ | $\frac{1}{2} \times (\text{DPC}(h, Y, S) + \text{DNC}(h, Y, S))$ |

distributions $P$ and $Q$ are the same. MMD has been used to boost the power of two-sample tests Chatterjee and Bhattacharya [2025]. We provide the formal definitions of RKHS and MMD, based on which our metric $\text{EO}_k$ is built.

**Definition 4.1** (Reproducing Kernel Hilbert Space; RKHS). *A Hilbert space, $\mathcal{H}$, containing functions mapping from a set $\mathcal{X}$ is called a* Reproducing Kernel Hilbert Space (RKHS) *if, $\forall x \in \mathcal{X}$, there exists a function $\varphi_x \in \mathcal{H}$, such that $\forall f \in \mathcal{H}$, we have*

$$\langle f, \varphi_x \rangle_{\mathcal{H}} = f(x),$$

*where $\langle \cdot, \cdot \rangle_{\mathcal{H}}$ is the inner product in space $\mathcal{H}$. The* reproducing kernel $k : \mathcal{X} \times \mathcal{X} \mapsto \mathbb{R}$ *of $\mathcal{H}$ is defined as follows:*

$$k(x, y) = \langle \varphi_x, \varphi_y \rangle_{\mathcal{H}}, \, \forall x, y \in \mathcal{X}.$$

Notably, given a pair of random vectors, MMD reflects the RKHS norm of between the kernel mean embeddings of the involved random vectors, that is,

**Definition 4.2** (Kernel Mean Embedding, Sriperumbudur et al. [2011], p. 2390). *Let $\mathcal{H}_k$ be an RKHS containing functions mapping from $\mathcal{X}$ to $\mathbb{R}$, and $k : \mathcal{X} \times \mathcal{X} \mapsto \mathbb{R}$ be the corresponding reproducing kernel. Given the set of all Borel probability measures defined on the topological space $\mathcal{X}$, a measurable and bounded kernel, $k$, is said to be* characteristic *if the following mapping is injective:*

$$P \mapsto \mu_P := \int_{\mathcal{X}} k(\cdot, x) dP(x).$$

*Here, $\mu_P \in \mathcal{H}$ is called the* kernel mean embedding *of distribution $P$.*

**Definition 4.3** (Maximum Mean Discrepancy; MMD, Gretton et al. [2012], Lemma 4). *Let $X$ and $Y$ be random vectors taking values in $\mathcal{X}$. Given a reproducing kernel $k : \mathcal{X} \times \mathcal{X} \mapsto \mathbb{R}$, the* Maximum Mean Discrepancy (MMD) *between $X$ and $Y$ is defined as the* Integral Probability Metric (IPM) *induced by the unit ball of the associated reproducing kernel Hilbert space $\mathcal{H}$. Formally,*

$$\gamma_k(X, Y) = \sup_{\|f\|_{\mathcal{H}} \leq 1} |\mathbb{E}[f(X)] - \mathbb{E}[f(Y)]|,$$

*where $\| \cdot \|_{\mathcal{H}}$ denotes the norm in $\mathcal{H}$. Equivalently, let $\mu_X := \mathbb{E}[k(\cdot, X)]$ and $\mu_Y := \mathbb{E}[k(\cdot, Y)]$ be the kernel mean embeddings of $X$ and $Y$, respectively. We have*

$$\gamma_k(X, Y) = \|\mu_X - \mu_Y\|_{\mathcal{H}}.$$

*In cases where $k$ is a characteristic kernel as defined in Definition 4.2, $\gamma_k(X, Y) = 0$ if and only if the involved probability distributions are the same.*

**Theorem 4.4** (Upper bound of $\gamma_k$ via TVD, Sriperumbudur et al. [2010b]). *Assume $\sup_x k(x, x) \leq \nu < \infty$, where $k$ is measurable on $M$. Then for any probability metrics $P$ and $Q$ embedded in set $M$, let $X \sim P$ and $Y \sim Q$, we have*

$$\gamma_k(X, Y) \leq 2\sqrt{\nu} d_{\text{TV}}(X, Y),$$

*where $d_{\text{TV}}$ denotes the TVD.*

## 4.2 Proof of Lemma 2.2

*Proof of Lemma 2.2.* Given the specified reproducing kernel $k$ as mentioned in the above lemma, let $\mathcal{H}$ be the corresponding reproducing kernel Hilbert space. Suppose $h \in \mathcal{H}$ and $\|h\|_{\mathcal{H}} \leq \nu^{-1/2}$, let $\langle \cdot, \cdot \rangle_{\mathcal{H}}$ be the inner product defined for the RKHS $\mathcal{H}$, and $\| \cdot \|_{\mathcal{H}}$ be the norm defined based on the inner product. Then, based on the definition of reproducing kernel, we have

$$
\begin{aligned}
\|h\|_\infty \quad &= \quad \sup_{z \in \mathbb{R}^{|Z|}} |h(z)| \\
&= \quad \sup_{z \in \mathbb{R}^{|Z|}} |\langle h, k(z, \cdot) \rangle_{\mathcal{H}}| \\
&\overset{\text{Cauchy Schwarz's}}{\leq} \quad \sup_{z \in \mathbb{R}^{|Z|}} \|h\|_{\mathcal{H}} \|k(z, \cdot)\|_{\mathcal{H}} \\
&= \quad \|h\|_{\mathcal{H}} \sup_{z \in \mathbb{R}^{|Z|}} \sqrt{k(z, z)} \\
&\overset{\sup_{z \in \mathbb{R}^{|Z|}} k(z,z) \leq \nu}{\leq} \quad \|h\|_{\mathcal{H}} \sqrt{\nu} \\
&\leq \quad 1.
\end{aligned}
$$

That is, the map of $h$ is contained within $[-1, 1]$. Consequently, the map of $(h+1)/2$ is contained within $[0, 1]$. $\qquad\square$

## 4.3 Proof of Lemma 2.4

*Proof of Lemma 2.4.* Suppose $\gamma_k(Z_0, Z_1) \leq \alpha$, then, based on Definition 4.3, we have

$$
\sup_{\|h\|_{\mathcal{H}} \leq 1} |\mathbb{E}[h(Z_0)] - \mathbb{E}[h(Z_1)]| \leq \alpha.
$$

Consequently, we have

$$
\sup_{\|h\|_{\mathcal{H}} \leq \nu^{-1/2}} |\mathbb{E}[h(Z_0)] - \mathbb{E}[h(Z_1)]| \leq \nu^{-1/2}\alpha,
$$

$$
\Leftrightarrow \sup_{\|h\|_{\mathcal{H}} \leq \nu^{-1/2}} \left| \mathbb{E}\left[\frac{h+1}{2}(Z_0)\right] - \mathbb{E}\left[\frac{h+1}{2}(Z_1)\right] \right| \leq \frac{\nu^{-1/2}}{2}\alpha,
$$

That is,

$$
|\mathbb{E}[g(Z_0)] - \mathbb{E}[g(Z_1)]| \leq \frac{\nu^{-1/2}\alpha}{2}, \quad \forall g \in \mathcal{G}.
$$

Since $\widehat{S} \sim \text{Bernoulli}(g(Z))$, we have

$$
\left| \Pr(\widehat{S} = 1 | S = 0) - \Pr(\widehat{S} = 1 | S = 1) \right| \leq \frac{\nu^{-1/2}\alpha}{2}, \quad \forall g \in \mathcal{G}.
$$

Let $\widehat{S}^*$ be the optimal predictor with respective to the sensitive attribute $S$, based on the set $\mathcal{G}$, then we have

$$
\left| \Pr(\widehat{S}^* = 1 | S = 0) - \Pr(\widehat{S}^* = 1 | S = 1) \right| = \left| \Pr(\widehat{S}^* = 1 | S = 1) + \Pr(\widehat{S}^* = 0 | S = 0) - 1 \right|
$$

$$
= \left| 2 \sup_{h \in \mathcal{H}} \text{BA}(h; Z, S) - 1 \right|.
$$

Consequently, $\sup_{h \in \mathcal{H}} \text{BA}(h; Z, S) \leq (2 + \nu^{-1/2}\alpha)/4$.

On the other hand, suppose the representation $Z$ is $\beta$-discriminative with respect to MMD, then we have $\gamma_k(Z^0, Z^1) \geq \beta$. Based on Theorem 4.4, we have $d_{\text{TV}}(Z^0, Z^1) \geq \frac{1}{2}\nu^{-1/2}\beta$. Then, based on [Shen et al., 2022, Proposition 2], we have $\sup_{g:\mathcal{Z} \mapsto [0,1]} \text{BA}(h; Z, Y) \geq \frac{2 + \nu^{-1/2}\beta}{4}$. Moreover, since $d_{\text{TV}} \leq 1$, we have $\beta \leq 2\nu^{1/2}$. $\qquad\square$

## 4.4 Proof of Theorem 2.5

*Proof of Theorem 2.5.* Suppose the feasible set of the classifiers is given as the function set $\mathcal{G}$ as defined in Lemma 2.4, i.e., $\mathcal{G} = \{g = (h+1)/2 \mid h \in \mathcal{H}, \|h\|_{\mathcal{H}} \leq \nu^{-1/2}\}$, where $((h+1)/2)(z) = (h(z)+1)/2, \forall z$, then we can rewrite the possible maximal value of the involved fairness notions as follows:

**Independence.** To begin with, based on Bayes' law, $Z_s = \Pr(Y = 0 \mid S = s)Z_s^0 + \Pr(Y = 1 \mid S = s)Z_s^1$, $s = 0, 1$.

$$
\begin{aligned}
\sup_{h \in \mathcal{H}} \mathrm{DP}(\widehat{Y}_g, S) &= \sup_{h \in \mathcal{H}} \left| \Pr(\widehat{Y}_g = 1 \mid S = 1) - \Pr(\widehat{Y}_g = 1 \mid S = 0) \right| \\
&= \sup_{h \in \mathcal{H}} \left| \mathbb{E}[\widehat{Y}_g \mid S = 1] - \mathbb{E}[\widehat{Y}_g \mid S = 0] \right| \\
&= \sup_{h \in \mathcal{H}} \left| \mathbb{E}[g(Z_1)] - \mathbb{E}[g(Z_0)] \right| \\
&= \sup_{\|h\|_{\mathcal{H}} \leq \nu^{-1/2}} \left| \mathbb{E}\left[ \frac{h+1}{2}(Z_1) \right] - \mathbb{E}\left[ \frac{h+1}{2}(Z_0) \right] \right| \\
&= \frac{1}{2} \sup_{\|h\|_{\mathcal{H}} \leq \nu^{-1/2}} \left| \mathbb{E}[h(Z_1)] - \mathbb{E}[h(Z_0)] \right| \\
&= (2\nu^{1/2})^{-1} \sup_{\|h\|_{\mathcal{H}} \leq 1} \left| \mathbb{E}[h(Z_1)] - \mathbb{E}[h(Z_0)] \right| \\
&= (2\nu^{1/2})^{-1} \gamma_k(Z_0, Z_1).
\end{aligned}
$$

**Separation.**

$$
\begin{aligned}
\sup_{h \in \mathcal{H}} \mathrm{DOpp}(h; Z, Y, S) &= \sup_{h \in \mathcal{H}} \left| \Pr(\widehat{Y} = 1 \mid Y = 1, S = 1) - \Pr(\widehat{Y} = 1 \mid Y = 1, S = 0) \right| \\
&= \sup_{h \in \mathcal{H}} \left| \mathbb{E}\left[ g(Z_1^1) \right] - \mathbb{E}\left[ g(Z_0^1) \right] \right| \\
&= \frac{1}{2} \sup_{\|h\|_{\mathcal{H}} \leq \nu^{-1/2}} \left| \mathbb{E}\left[ h(Z_1^1) \right] - \mathbb{E}\left[ h(Z_0^1) \right] \right| \\
&= (2\nu^{1/2})^{-1} \gamma_k(Z_1^1, Z_0^1).
\end{aligned}
$$

Similarly, we have $\sup_{h \in \mathcal{H}} \mathrm{DR}(h; Z, Y, S) = (2\nu^{1/2})^{-1} \gamma_k(Z_1^0, Z_0^1)$. Based on the upper bounds for $\sup_{h \in \mathcal{H}} \mathrm{DOpp}$ and $\sup_{h \in \mathcal{H}} \mathrm{DR}$, we have $\sup_{h \in \mathcal{H}} \mathrm{DOdds} \leq \sup_{h \in \mathcal{H}} \mathrm{DOpp} + \sup_{h \in \mathcal{H}} \mathrm{DR} = (2\nu^{1/2})^{-1}(\gamma_k(Z_1^1, Z_0^1) + \gamma_k(Z_1^0, Z_0^0))$.

**Lower and upper bounds.** Recall Definition 4.2, we have

$$
\begin{aligned}
& \|\mu_0 - \mu_1\|_{\mathcal{H}} \\
=~& \left\| p_{0|0}\mu_0^0 + p_{1|0}\mu_0^1 - p_{0|1}\mu_1^0 - p_{1|1}\mu_1^1 \right\|_{\mathcal{H}} \\
=~& \left\| p_{0|0}\mu_0^0 + (1 - p_{0|0})\mu_0^1 - p_{0|1}\mu_1^0 - (1 - p_{0|1})\mu_1^1 \right\|_{\mathcal{H}} \\
=~& \left\| p_{0|0}(\mu_0^0 - \mu_1^0) + (1 - p_{0|0})(\mu_0^1 - \mu_1^1) + (p_{0|0} - p_{0|1})(\mu_1^0 - \mu_1^1) \right\|_{\mathcal{H}}.
\end{aligned}
$$

In cases where $Y \perp\!\!\!\perp S$, we have $p_{0|0} = p_{0|1}$. Consequently, $\|\mu_0 - \mu_1\|_{\mathcal{H}} = \left\| p_{0|0}(\mu_0^0 - \mu_1^0) + (1 - p_{0|0})(\mu_0^1 - \mu_1^1) \right\|_{\mathcal{H}}$. Based on the triangle's inequality, we have

$\|\mu_0 - \mu_1\|_{\mathcal{H}} \geq \left| p_{0|0}\gamma_k(Z_0^0, Z_1^0) - p_{1|0}\gamma_k(Z_0^1, Z_1^1) \right|$, and $\|\mu_0 - \mu_1\|_{\mathcal{H}} \leq p_{0|0}\gamma_k(Z_0^0, Z_1^0) + p_{1|0}\gamma_k(Z_0^1, Z_1^1)$.

Suppose $p_{0|0} \neq p_{0|1}$, we have

$$
\|\mu_0 - \mu_1\|_{\mathcal{H}} \geq \left| \left\| p_{0|0}(\mu_0^0 - \mu_1^0) + (1 - p_{0|0})(\mu_0^1 - \mu_1^1) \right\|_{\mathcal{H}} - |p_{0|0} - p_{0|1}| \left\| \mu_1^0 - \mu_1^1 \right\|_{\mathcal{H}} \right|.
$$

$\square$

## 4.5 Proof of Theorem 2.6

*Proof of Theorem 2.6.* We start with the following lemma, whose proof is referred to Appendix 4.6.

**Lemma 4.5** (Calibration and TVD). *Considering the calibration notions, i.e, DPC, DNC, and DC, proposed in Shen et al. [2022] (Table 1). We have*

$$d_{\mathrm{TV}}\left((Y, h(Z)) \mid S = 0, (Y, h(Z)) \mid S = 1\right) = 2\,\mathrm{DC}(h, Y, S),$$
$$d_{\mathrm{TV}}\left((Y, h(Z)) \mid S = 0, (Y, h(Z)) \mid S = 1\right) \geq \mathrm{DPC}(h, Y, S), \textit{ and}$$
$$d_{\mathrm{TV}}\left((Y, h(Z)) \mid S = 0, (Y, h(Z)) \mid S = 1\right) \geq \mathrm{DNC}(h, Y, S).$$

Return to the proof of Theorem 2.6. Since $\sup_u k_{[0,1]}(u, u) \leq \nu_{[0,1]}$ and $\sup_y k_{\{0,1\}}(y, y) \leq \nu_{\{0,1\}}$, we have

$$
\begin{aligned}
\sup_{(u,y)} k_{[0,1]} \otimes k_{\{0,1\}}((u, y), (u, y)) &= \sup_{(u,y)} k_{[0,1]}(u, u) k_{\{0,1\}}(y, y) \\
&= \sup_u k_{[0,1]}(u, u) \sup_y k_{\{0,1\}}(y, y) \\
&= \nu_{[0,1]} \nu_{\{0,1\}}.
\end{aligned}
$$

Note that $\mathrm{DC}(h, Y, S)) = \frac{1}{2} d_{\mathrm{TV}}\left((Y, h(Z)) \mid S = 0, (Y, h(Z)) \mid S = 1\right)$. Based on Theorem 4.4, we have

$$
\begin{aligned}
&d_{\mathrm{TV}}\left((Y, h(Z)) \mid S = 0, (Y, h(Z)) \mid S = 1\right) \\
&\geq \frac{1}{2}(\nu_{[0,1]}\nu_{\{0,1\}})^{-1/2} \gamma_{k_{[0,1]} \otimes k_{\{0,1\}}}((Y, h(Z)) \mid S = 0, (Y, h(Z)) \mid S = 1).
\end{aligned}
$$

Consequently, we have

$$\mathrm{DC}(h, Y, S)) \geq \frac{1}{4}(\nu_{[0,1]}\nu_{\{0,1\}})^{-1/2} \gamma_{k_{[0,1]} \otimes k_{\{0,1\}}}((Y, h(Z)) \mid S = 0, (Y, h(Z)) \mid S = 1).$$

From the definition of MMD (Definition 4.3), we have

$$
\begin{aligned}
&\gamma_{k_{[0,1]} \otimes k_{\{0,1\}}}((Y, h(Z)) \mid S = 0, (Y, h(Z)) \mid S = 1) \\
&= \sup_{\|f\|_{\mathcal{H}_{[0,1]} \otimes \mathcal{H}_{\{0,1\}}} \leq 1} |\mathbb{E}[f(Y, h(Z)) \mid S = 0] - \mathbb{E}[f(Y, h(Z)) \mid S = 1]|.
\end{aligned}
$$

Consider the function set $\mathcal{F} := \left\{ f \mid f(y, u) = \tilde{g}(u), \|\tilde{g}\|_{\mathcal{H}_{[0,1]}} \leq 1 \right\}$. Let $\mathbf{1}$ be a constant function that maps every input to 1. $\forall f \in \mathcal{F}$, based on the property of the tensor product RKHS Szabó and Sriperumbudur [2018], we have

$$
\begin{aligned}
\|f\|_{\mathcal{H}_{[0,1]} \otimes \mathcal{H}_{\{0,1\}}} &= \|\mathbf{1}\|_{\mathcal{H}_{\{0,1\}}} \|\tilde{g}\|_{\mathcal{H}_{[0,1]}} \\
&\leq \|\mathbf{1}\|_{\mathcal{H}_{\{0,1\}}}.
\end{aligned}
$$

Note that $\{0, 1\}$ is a finite space. Let $\mathbf{K} := (k_{\{0,1\}}(i, j))_{i,j \in \{0,1\}}$ be the Gram's matrix. Consider the vector $\boldsymbol{\alpha} = (\alpha_1, \alpha_2)^T$, s.t., $\mathbf{1} = (k_{\{0,1\}}(0, \cdot), k_{\{0,1\}}(1, \cdot))\boldsymbol{\alpha}$. It can be observed that $\mathbf{K}\boldsymbol{\alpha} = (1, 1)^T$. For simplicity, denote $k_{\{0,1\}}(i, j)$ as $k_{i,j}$. We have

$$
\begin{aligned}
\|\mathbf{1}\|_{\mathcal{H}_{\{0,1\}}} &= \sqrt{\boldsymbol{\alpha}^T \mathbf{K} \boldsymbol{\alpha}} \\
&= \sqrt{(1, 1)\mathbf{K}^{-1}(1, 1)^T} \\
&= \sqrt{\frac{k_{0,0} + k_{1,1} - 2k_{0,1}}{|\mathbf{K}|}} \\
&= 1.
\end{aligned}
$$

Consequently, we have $\|f\|_{\mathcal{H}_{[0,1]} \otimes \mathcal{H}_{\{0,1\}}} \leq 1, \forall f \in \mathcal{F}$. That is, $\mathcal{F} \subset \{h \mid \|h\|_{\mathcal{H}_{[0,1]} \otimes \mathcal{H}_{\{0,1\}}} \leq 1\}$. Combining the above results, we have

$$
\begin{aligned}
&\gamma_{k_{[0,1]} \otimes k_{\{0,1\}}}((Y, h(Z)) \mid S = 0, (Y, h(Z)) \mid S = 1) \\
&\geq \sup_{f \in \mathcal{F}} |\mathbb{E}[f(Y, h(Z)) \mid S = 0] - \mathbb{E}[f(Y, h(Z)) \mid S = 1]| \\
&= \sup_{\|\tilde{g}\|_{\mathcal{H}_{[0,1]}} \leq 1} |\mathbb{E}[\tilde{g} \circ h(Z) \mid S = 0] - \mathbb{E}[\tilde{g} \circ h(Z) \mid S = 1]|.
\end{aligned}
$$

Let id be the identity map embedded in $[0, 1]$. Suppose $\mathrm{id} \in \mathcal{H}_{[0,1]}$. Considering every possible predicting function $h \in \mathcal{G}$, we have

$$\sup_{\|\tilde{g}\|_{\mathcal{H}_{[0,1]}} \leq 1, h \in \mathcal{G}} |\mathbb{E}[\tilde{g} \circ h(Z) \mid S = 0] - \mathbb{E}[\tilde{g} \circ h(Z) \mid S = 1]|$$

$$= \sup_{\|\tilde{g}\|_{\mathcal{H}_{[0,1]}} \leq 1, \|h\|_{\mathcal{H}} \leq \nu^{-1/2}} |\mathbb{E}[\tilde{g}((h(Z)+1)/2) \mid S = 0] - \mathbb{E}[\tilde{g}((h(Z)+1)/2) \mid S = 1]|$$

$$= \|\mathrm{id}\|_{\mathcal{H}_{[0,1]}}^{-1} \sup_{\|\tilde{g}\|_{\mathcal{H}_{[0,1]}} \leq \|\mathrm{id}\|_{\mathcal{H}_{[0,1]}}, \|h\|_{\mathcal{H}} \leq \nu^{-1/2}} |\mathbb{E}[\tilde{g}((h(Z)+1)/2) \mid S = 0] - \mathbb{E}[\tilde{g}((h(Z)+1)/2) \mid S = 1]|$$

$$\geq \|\mathrm{id}\|_{\mathcal{H}_{[0,1]}}^{-1} \sup_{\|h\|_{\mathcal{H}} \leq \nu^{-1/2}} |\mathbb{E}[\mathrm{id}((h(Z)+1)/2) \mid S = 0] - \mathbb{E}[\mathrm{id}((h(Z)+1)/2) \mid S = 1]|$$

$$= (2\|\mathrm{id}\|_{\mathcal{H}_{[0,1]}})^{-1} \sup_{\|h\|_{\mathcal{H}} \leq \nu^{-1/2}} |\mathbb{E}[h(Z) \mid S = 0] - \mathbb{E}[h(Z) \mid S = 1]|$$

$$= (2\|\mathrm{id}\|_{\mathcal{H}_{[0,1]}})^{-1} \nu^{-1/2} \sup_{\|h\|_{\mathcal{H}} \leq 1} |\mathbb{E}[h(Z) \mid S = 0] - \mathbb{E}[h(Z) \mid S = 1]|$$

$$= (2\|\mathrm{id}\|_{\mathcal{H}_{[0,1]}})^{-1} \nu^{-1/2} \gamma_k(Z_0, Z_1)$$

$$= \|\mathrm{id}\|_{\mathcal{H}_{[0,1]}}^{-1} \sup_{h \in \mathcal{H}} \mathrm{DP}(h; Z, Y, S).$$

$\square$

### 4.6 Proof of Lemma 4.5

*Proof of Lemma 4.5.* Recall the notions for calibration, i.e., DPC, DNC, and DC, proposed in Shen et al. [2022] (Table 1):

$$\mathrm{DPC}(h, Y, S) = \frac{1}{2} \sum_{t \in [0,1]} |\Pr(Y = 1, h(Z) = t \mid S = 1) - \Pr(Y = 1, h(Z) = t \mid S = 0)|,$$

$$\mathrm{DNC}(h, Y, S) = \frac{1}{2} \sum_{t \in [0,1]} |\Pr(Y = 0, h(Z) = t \mid S = 1) - \Pr(Y = 0, h(Z) = t \mid S = 0)|,$$

$$\mathrm{DC}(h, Y, S) = \frac{1}{2} (\mathrm{DPC}(h, Y, S) + \mathrm{DNC}(h, Y, S)).$$

Based on the expressions listed above, we have

$$d_{\mathrm{TV}}((Y, h(Z)) \mid S = 0, (Y, h(Z)) \mid S = 1)$$
$$= \frac{1}{2} \sum_{t \in [0,1]} [|\Pr(Y = 1, h(Z) = t \mid S = 0) - \Pr(Y = 1, h(Z) = t \mid S = 1)|$$
$$+ |\Pr(Y = 0, h(Z) = t \mid S = 0) - \Pr(Y = 0, h(Z) = t \mid S = 1)|]$$
$$= \mathrm{DPC}(h, Y, S) + \mathrm{DNC}(h, Y, S)$$
$$= 2\,\mathrm{DC}(h, Y, S).$$

Since $2\,\mathrm{DC} = \mathrm{DPC} + \mathrm{DNC}$, the next two inequalities can be derived. $\square$

### 4.7 Supplementary for Gaussian Complexity

In the following, we provide a concentration inequality for the Gaussian complexity, revealing that it can be empirically estimated from a realization, simplifying its computation.

**Proposition 4.6** (Concentration of Gaussian Complexity). *Given a function class $\mathcal{F} := \{h : \mathcal{X} \mapsto \mathcal{Z}\}$, where $\mathcal{Z} \subseteq \mathbb{R}^d$. Denote the set $\{f(x) \mid f \in \mathcal{F}, x \in \mathcal{X}\}$ in $\mathcal{Z}$ as $\mathcal{F}(\mathcal{X})$. Let $\mathcal{F}(\mathbf{X})$ be a set in $\mathcal{Z}^n$ defined as $\mathcal{F}(\mathbf{X}) := \{(f(X_1), \ldots, f(X_n)) \mid f \in \mathcal{F}\}$. Let $D(\mathcal{F}(\mathcal{X})) := \sup_{z, z' \in \mathcal{F}(\mathcal{X})} \|z - z'\|$ be the Euclidean width of the set $\mathcal{F}(\mathcal{X})$, $G_{nd}(\mathcal{F}(\mathbf{X}))$ be the Gaussian complexity of the set $\mathcal{F}(\mathbf{X})$ as defined in (14). Then $\forall \delta \in (0, 1)$, with probability at least $1 - \delta$, we have*

$$\left| \sup_{f \in \mathcal{F}} \sum_{i=1}^n \langle \xi_i, f(X_i) \rangle - G_{nd}(\mathcal{F}(\mathbf{X})) \right| \leq D(\mathcal{F}(\mathcal{X})) \sqrt{n \log\left(\frac{2}{\delta}\right)},$$

where $\xi_i \overset{i.i.d.}{\sim} \mathcal{N}(\mathbf{0}_d, I_d)$, $\forall i$. Moreover, we have

$$|G_{nd}(\mathcal{F}(\mathbf{X})) - \mathbb{E}[G_{nd}(\mathcal{F}(\mathbf{X}))]| \leq D(\mathcal{F}(\mathcal{X}))\sqrt{\frac{nd}{2}\log\left(\frac{2}{\delta}\right)}.$$

Considering the case where the feasible set of the encoders is formulated by neural networks, which is a common scenario in FRL, its covering number is finite, as shown in Theorem 2 of Shen [2023], corresponding to the satisfaction of Assumption 2.7. In the following, we provide an upper bound for its Gaussian complexity in the following proposition.

**Proposition 4.7** (Gaussian Complexities of Feed-Forward Neural Networks, Ni and Huo [2024], Proposition 18). *Consider a feed-forward neural network with depth $\iota$, which is given by the function $f_{nn}^\iota : \mathbb{R}^d \mapsto \mathbb{R}$ defined as follows*

$$f_{nn}^{(\iota)}(x) := l^{(\iota)} \circ \cdots \circ l^{(1)}(x) \equiv l^{(\iota)}\left(\cdots l^{(2)}\left(l^{(1)}(x)\right)\cdots\right),$$

*where $d_0 = d$, $d_\iota = 1$, and $l^{(\iota)} := W^{(\iota)}x$ for a specified matrix $\mathbb{R}^{1 \times d_{\iota-1}}$. Here, for $k = 1, \ldots, \iota - 1$, $l^{(k)} : \mathbb{R}^{d_{k-1}} \mapsto \mathbb{R}^{d_k}$ is the k-th hidden layer consists of a coordinate-wise composition of an activation function $\sigma : \mathbb{R} \mapsto \mathbb{R}$ and an affine map, namely, $l^{(k)}(x) := \phi(W^{(k)}x)$ for an given interaction matrix $W^{(k)} \in \mathbb{R}^{d_k \times d_{k-1}}$. Let the interaction matrices be the parameters to be tuned, the corresponding class of neural networks is given as follows:*

$$\mathcal{F} := \left\{ f_{nn}^{(\iota)}(x) \,\middle|\, \left\|W^{(k)}\right\|_{1,\infty} \leq \omega, \forall k \right\},$$

*where for a given matrix $W$, the $\|\cdot\|_{1,\infty}$ norm is defined as $\|W\|_{1,\infty} = \max_i \sum_j |W_{i,j}|$. Suppose the activation function $\sigma$ is $\lambda$-Lipschitz, let $\mathbf{X} := (X_1, \ldots, X_n)^T \in \mathbb{R}^{n \times d_0}$, we have*

$$\mathcal{G}(\mathcal{F}(\mathbf{X})) \leq (2\omega)^\iota \lambda^{\iota-1} \sqrt{2\log(2d_0)} \max_k \sqrt{\sum_{i=1}^n X_{i,k}^2}.$$


## 10 Broader impacts

Question: Does the paper discuss both potential positive societal impacts and negative societal impacts of the work performed?

Answer: [Yes]

Justification: This paper addresses fair representation learning, a topic inherently tied to the social impact of machine learning systems. The proposed fairness constraint promotes equitable outcomes by ensuring that, conditional on the target variable, model predictions are independent of sensitive attributes. This directly contributes to mitigating discriminatory behavior in downstream applications.

Guidelines:

- The answer NA means that there is no societal impact of the work performed.
- If the authors answer NA or No, they should explain why their work has no societal impact or why the paper does not address societal impact.
- Examples of negative societal impacts include potential malicious or unintended uses (e.g., disinformation, generating fake profiles, surveillance), fairness considerations (e.g., deployment of technologies that could make decisions that unfairly impact specific groups), privacy considerations, and security considerations.
- The conference expects that many papers will be foundational research and not tied to particular applications, let alone deployments. However, if there is a direct path to any negative applications, the authors should point it out. For example, it is legitimate to point out that an improvement in the quality of generative models could be used to generate deepfakes for disinformation. On the other hand, it is not needed to point out that a generic algorithm for optimizing neural networks could enable people to train models that generate Deepfakes faster.
- The authors should consider possible harms that could arise when the technology is being used as intended and functioning correctly, harms that could arise when the technology is being used as intended but gives incorrect results, and harms following from (intentional or unintentional) misuse of the technology.
- If there are negative societal impacts, the authors could also discuss possible mitigation strategies (e.g., gated release of models, providing defenses in addition to attacks, mechanisms for monitoring misuse, mechanisms to monitor how a system learns from feedback over time, improving the efficiency and accessibility of ML).

## 11 Safeguards

Question: Does the paper describe safeguards that have been put in place for responsible release of data or models that have a high risk for misuse (e.g., pretrained language models, image generators, or scraped datasets)?

Answer: [NA]

Justification: This paper poses no such risks.

Guidelines:

- The answer NA means that the paper poses no such risks.
- Released models that have a high risk for misuse or dual-use should be released with necessary safeguards to allow for controlled use of the model, for example by requiring that users adhere to usage guidelines or restrictions to access the model or implementing safety filters.

- Datasets that have been scraped from the Internet could pose safety risks. The authors should describe how they avoided releasing unsafe images.
- We recognize that providing effective safeguards is challenging, and many papers do not require this, but we encourage authors to take this into account and make a best faith effort.

12. **Licenses for existing assets**

Question: Are the creators or original owners of assets (e.g., code, data, models), used in the paper, properly credited and are the license and terms of use explicitly mentioned and properly respected?

Answer: [NA]

Justification: This paper does not use existing assets.

Guidelines:

- The answer NA means that the paper does not use existing assets.
- The authors should cite the original paper that produced the code package or dataset.
- The authors should state which version of the asset is used and, if possible, include a URL.
- The name of the license (e.g., CC-BY 4.0) should be included for each asset.
- For scraped data from a particular source (e.g., website), the copyright and terms of service of that source should be provided.
- If assets are released, the license, copyright information, and terms of use in the package should be provided. For popular datasets, `paperswithcode.com/datasets` has curated licenses for some datasets. Their licensing guide can help determine the license of a dataset.
- For existing datasets that are re-packaged, both the original license and the license of the derived asset (if it has changed) should be provided.
- If this information is not available online, the authors are encouraged to reach out to the asset's creators.

13. **New assets**

Question: Are new assets introduced in the paper well documented and is the documentation provided alongside the assets?

Answer: [NA]

Justification: This paper does not release new assets.

Guidelines:

- The answer NA means that the paper does not release new assets.
- Researchers should communicate the details of the dataset/code/model as part of their submissions via structured templates. This includes details about training, license, limitations, etc.
- The paper should discuss whether and how consent was obtained from people whose asset is used.
- At submission time, remember to anonymize your assets (if applicable). You can either create an anonymized URL or include an anonymized zip file.

14. **Crowdsourcing and research with human subjects**

Question: For crowdsourcing experiments and research with human subjects, does the paper include the full text of instructions given to participants and screenshots, if applicable, as well as details about compensation (if any)?

Answer: [NA]

Justification: This paper does not involve crowdsourcing nor research with human subjects.

Guidelines:

- The answer NA means that the paper does not involve crowdsourcing nor research with human subjects.
- Including this information in the supplemental material is fine, but if the main contribution of the paper involves human subjects, then as much detail as possible should be included in the main paper.
- According to the NeurIPS Code of Ethics, workers involved in data collection, curation, or other labor should be paid at least the minimum wage in the country of the data collector.

### 15 Institutional review board (IRB) approvals or equivalent for research with human subjects

Question: Does the paper describe potential risks incurred by study participants, whether such risks were disclosed to the subjects, and whether Institutional Review Board (IRB) approvals (or an equivalent approval/review based on the requirements of your country or institution) were obtained?

Answer: [NA]

Justification: This paper does not involve crowdsourcing nor research with human subjects.

Guidelines:

- The answer NA means that the paper does not involve crowdsourcing nor research with human subjects.
- Depending on the country in which research is conducted, IRB approval (or equivalent) may be required for any human subjects research. If you obtained IRB approval, you should clearly state this in the paper.
- We recognize that the procedures for this may vary significantly between institutions and locations, and we expect authors to adhere to the NeurIPS Code of Ethics and the guidelines for their institution.
- For initial submissions, do not include any information that would break anonymity (if applicable), such as the institution conducting the review.

### 16 Declaration of LLM usage

Question: Does the paper describe the usage of LLMs if it is an important, original, or non-standard component of the core methods in this research? Note that if the LLM is used only for writing, editing, or formatting purposes and does not impact the core methodology, scientific rigorousness, or originality of the research, declaration is not required.

Answer: [NA]

Justification: The core method development in this research does not involve LLMs as any important, original, or non-standard components.

Guidelines:

- The answer NA means that the core method development in this research does not involve LLMs as any important, original, or non-standard components.
- Please refer to our LLM policy (`https://neurips.cc/Conferences/2025/LLM`) for what should or should not be described.

