# OpenReview forum: "Kernel-based Equalized Odds: A Quantification of Accuracy-Fairness Trade-off in Fair Representation Learning"
_NeurIPS.cc/2025/Conference — NeurIPS 2025 poster_

### Official Review · Reviewer_4aTM · 2025-06-03

**Clarity:** 3
**Significance:** 3
**Originality:** 2
**Rating:** 4
**Confidence:** 3

**Summary:**

This paper proposes a kernel-based Equalized Odds (EQ) criterion for characterization fairness in machine learning. An empirical estimator is provided to estimate the kernel EQ, and the author establishes theoretical guarantees for the estimator as well as fairness-accuracy tradeoffs.

**Questions:**

Does kernel misspecification lead to false fairness claims using the proposed criterion?

**Ethical Concerns:**

["NO or VERY MINOR ethics concerns only"]

**Final Justification:**

The authors address my concern and I maintain my score.

**Quality:**

3

**Strengths And Weaknesses:**

## Strengths
  - The proposed kernel-based EQ could be a useful extension of EQ with robust theoretical guarantees.
  - An empirical estimator is provided for calculating the kernel EQ which is practical and scalable.
  - The authors show desirable theoretical guarantees for fairness-accuracy tradeoffs as well as convergence of the empirical estimator.

## Weaknesses

  - The proposed fairness criterion and the associated results are dependent on the given kernel. If the kernel is misspecified, the criterion may yield false fairness claims. Kernel misspecification could be less relevant for prediction tasks, but a concern for a criterion.
  - Kernel-based equalized odds and fairness-accuracy tradeoffs have already been studied in literature. Potential missing related work:
Tan et al.,  Learning Fair Representations for Kernel Models. AISTATS, 2020

---

> ### Author Rebuttal · Authors · 2025-07-28
>
> Thank you for your thoughtful review and for recognizing the technical soundness of our proposed framework, its theoretical guarantees, and its practical estimator. You raise two important technical points regarding kernel misspecification and the relationship to prior work by Tan et al. (2020). We welcome the opportunity to clarify both, as they are central to the contribution of our paper.
>
> ### **1. On the Robustness of Our Criterion to Kernel Choice**
>
> You raise an excellent and fundamental question: "Does kernel misspecification lead to false fairness claims using the proposed criterion?" This is a crucial point that we touch upon in **Lines 88-97**, where we discuss the expressiveness of the function class defined by the kernel. We appreciate the opportunity to elaborate on this, as we recognize our initial explanation may have been too concise.
>
> This concern is at the heart of all kernel-based distribution measures, and our framework relies on a powerful, standard resolution from the literature:
>
> - **The Role of Characteristic Kernels:** Our proposed metric, $\mathrm{EO}_k$, is a kernel-based Integral Probability Metric (IPM) that is defined using a **characteristic kernel** (e.g., Gaussian RBF), as is standard for MMD-based metrics. The established theory for these kernels (c.f., Sriperumbudur et al., 2010b) provides a strong guarantee: the corresponding Reproducing Kernel Hilbert Space (RKHS) is rich enough that the IPM will be zero *if and only if* the underlying distributions are identical. This is the core principle we allude to in **Lines 88-97** when discussing the richness of the unit ball of the RKHS.
> - **Implication for Fairness Claims:** Consequently, a "false fairness claim"—where our $\mathrm{EO}_k$ metric is zero but the conditional distributions of the representations are truly different—is theoretically avoided. With a sufficiently flexible characteristic kernel, whose parameters can be set via standard practices like the median heuristic, $\mathrm{EO}_k$ provides a reliable and robust measure of fairness violations.
>
> **Action:** We see now that this point is critical and deserves more prominence. In our revision, we will expand upon the discussion in Lines 88-97 to explicitly state the reliance on characteristic kernels to make this theoretical foundation unambiguous for the reader.
>
> ### **2. Clarifying Our Contribution Relative to Tan et al. (2020)**
>
> We appreciate you highlighting Tan et al. (2020). While it is an important paper on FRL for kernel models, our work addresses a fundamentally different research question and provides a complementary set of insights. The distinctions are crucial:
>
> 1. **Focus of Analysis (Fairness vs. Fairness):** Our primary contribution is a theoretical characterization of the **incompatibility between multiple fairness criteria** (EO, DP, and DC). We prove that under data bias, enforcing EO necessitates a quantifiable violation of the others. Tan et al., in contrast, analyze the trade-off between a single fairness notion and predictive accuracy.
> 2. **Nature of the Trade-off Analysis:** Our technical approach derives analytical lower bounds on these cross-metric fairness violations. Theirs is a geometric approach based on subspace projections. These are different and non-overlapping analytical tools that address different aspects of the fairness landscape.
> 3. **Core Problem Setting:** Most critically, our work operates in the standard group fairness setting where the sensitive attribute S is available at training time (a prerequisite for defining EO). Tan et al. address the distinct and challenging problem of learning fair representations **without access to S** at training time.
>
> **Action:** While both papers use kernel methods, they tackle different problems with different tools. Our paper's novelty lies in its distribution-level characterization of fairness-fairness trade-offs. We will revise our related work section to include a discussion of Tan et al. and articulate these clear distinctions.
>
> ------
>
> We hope this response clarifies these two important technical aspects of our work. We are confident that by incorporating these clarifications into the manuscript, the novelty and rigor of our contributions will be even more apparent. Thank you again for your constructive engagement.

---

> > ### Comment · Area_Chair_gXn7 · 2025-08-05
> > **kernel choice**
> >
> > Thanks to the authors for their detailed responses! I wanted to add that the use of a characteristic kernel is only theoretically sufficient to avoid problems in the case of infinite data (i.e., not in practice). Even a characteristic kernel, if poorly chosen, can make it unreliable to reason about IPMs: e.g., some types of difference between distributions can be emphasized while others are de-emphasized, so that we fail to notice a difference when there is one, or vice versa. An example might be choosing a wide Gaussian kernel when the corresponding distributions do not have a very smooth density: we would be good at noticing large-scale changes and bad at noticing small local shifts or high-frequency changes. (This could be an easy case to try out an example or two to see the practical effect.)

---

### Official Review · Reviewer_bR6s · 2025-06-24

**Clarity:** 2
**Significance:** 2
**Originality:** 3
**Rating:** 4
**Confidence:** 2

**Summary:**

This paper proposes a fairness measure for fair representation learning, which is a kernel-based formulation of EO ($EO_k$) and provides theoretical properties of this new proposed formulation. This new formulation uses MMD to measure the differences in representations between two demographic groups, enabling rigorous and interpretable quantification of independence, separation (via EO), and calibration within a unified framework. The main contribution of the paper is to quantify the trade-offs between these mutually incompatible fairness constraints and to analyze their implications.

**Questions:**

- While the paper emphasizes that $EO_k$ is not designed to optimize all fairness metrics simultaneously but rather to quantify the trade-offs between them, this conceptual distinction remains abstract without concrete justification. Can the authors provide a realistic case or application where such trade-off quantification is actually useful? For example, how would a practitioner interpret or act on the value of $EO_k$ in a model selection or deployment scenario? More generally, how does $EO_k$ support fairness-aware decision-making beyond what simpler metrics (e.g., separate TPR/FPR gaps or post-hoc fairness curves) already provide? What specific limitations of existing fairness criteria does $EO_k$ address that cannot be handled using DP or EO? i.e., why is a new kernel-based fairness metric necessary?

- In practical settings, fairness constraints are typically selected based on domain or specific requirements (e.g., some cases decision makers prefer equalized odds, some cases prefer accuracy parity), why is a unified definition preferable or more actionable?

- Using kernel-based definition will increase computational cost compared with DP and EO, so what practical gains justify this additional complexity?

**Ethical Concerns:**

["NO or VERY MINOR ethics concerns only"]

**Final Justification:**

I appreciate the authors’ detailed response, which addressed most of my questions. After also reading the comments from the other reviewers, I have decided to raise my score. I strongly encourage the authors to revise the paper as outlined in their rebuttal, particularly by restructuring it to make several key points more explicit. This will greatly enhance the paper’s clarity and accessibility for a broader audience. I have raised my score as a final justification of my initial evaluation.

**Limitations:**

The authors did not discuss the limitations of their approach. The main limitations are outlined in Weakness Point 1 and further elaborated in the questions I have raised.

**Quality:**

3

**Strengths And Weaknesses:**

Strengths:
- The new formulation is mathematically well-defined in a principled manner (Section 2.1), and the derivation of the generalization bounds and the closed-form empirical estimator is technically correct and solid.

Weaknesses:
- The main concern with this paper is the lack of empirical support (the submission contains no experiments, toy examples or any empirical validation). While the authors state that the work is theoretical, but provides a connection between theory and practice is still important. Without at least a simulated example demonstrating how $EO_k$ captures trade-offs in realistic scenarios, it's difficult to assess its practical significance.

- Section 2 presents the main theoretical contribution, and the section reads like a disconnected sequence of tehnical points (defs, lemmas, bounds), each introduced in isolation without sufficient transitions or motivation. For example. section 2.1 introduces the formulation of $EO_k$ and after that immediately moves to its RHKS form with no clear intuition or example to help the reader undersatnd the object being defined. Lemma 2.2 and 2.4 define technical function classes and bounds, but the role in the overall framework is not explained until much later.  Also, the notations are compact and mathematical shorthand that obscure the main ideas and reduce the readability. After reading Section 2, it is still unclear what the main theoretical message is beyond a set of bounds and definitions. The paper would benefit greatly from: (1) adding a motivating paragraph at the beginning of Section 2; (2) add some clear transitions between subsections to explain why each result is introduced and (3) maybe provide a summary paragraph at the end that synthesizes the theoretical results and explains their implications for fair representation learning.

- The necessity of using kernel-based MMD to formulate fairness constraints is not well justified. This design choice introduces additional complexity. In practice, simpler and more model-agnostic alternatives are often more interpretable and widely adopted.

- Similar to the above one, while the theoretical unification of multiple fairness definitions is interesting from a statistical perspective, in practice, practitioners typically select one or two fairness notions based on task requirements. The need to balance all three (independence, separation, and calibration) is not sufficiently motivated, and the paper lacks examples where such trade-offs arise naturally or meaningfully. In other words, the paper introduces $EO_k$ without sufficiently motivating the limitations of existing metrics or demonstrating that this new formulation enables better fairness–accuracy trade-offs in practical tasks. It remains unclear what real-world problems $EO_k$ solves that cannot already be addressed with existing fairness constraints.

---

> ### Author Rebuttal · Authors · 2025-07-28
>
> Thank you for your detailed review and for acknowledging the technical soundness of our formulation. Your feedback has been instrumental in helping us identify where the manuscript's presentation needs to be strengthened. Your core concerns about motivation, structure, and practical justification all point to a single root cause: our draft implicitly assumes familiarity with the context of recent Fair Representation Learning (FRL) research.
>
> To resolve the issues you raised, our main revision will be to restructure the paper around a clear "Problem-Solution" narrative, making this context explicit. Below, we outline how this new structure will directly address the points you raised.
>
> ### **1. Motivation and Structure (W1, W2, Q1)**
>
> You correctly noted that the paper currently lacks a strong motivational thread (W1) and that Section 2 can read as a disconnected set of technical points **(W2)**. This is a direct result of us not adequately setting the stage.
>
> The revised paper will resolve this by dedicating a new introduction paragraph upfront to the well-established problem our work solves:
>
> - **The Problem:** The machine learning community requires methods for *in-training fairness enforcement*. Post-hoc auditing metrics like TPR/FPR gaps (your **Q1**), while vital for evaluation, are non-differentiable and cannot be used directly in gradient-based optimization. This is the central motivation for FRL and, by extension, our work.
> - **The New Structure:** By establishing this "in-training vs. post-hoc" problem first, the technical details in Section 2 will no longer appear disconnected. Instead, they will be clearly framed as the necessary mathematical foundation for building our proposed solution. To further underscore the practical significance, we will also add a small-scale experiment demonstrating our method in a simulated application. The experimental results are referred to the rebuttal for Reviewer akC3.
>
> ### **2. Complexity and Computational Cost of metric $\mathrm{EO}_k$ (W3, Q3)**
>
> - You raised a crucial question about whether the complexity of our kernel-based method is justified over simpler alternatives and what its computational cost is **(W3, Q3)**.
>
>   Our revision will clarify that the relevant comparison is not to simple post-hoc metrics, but to other *in-training* enforcement methods. This context is key:
>
>   - **Superiority over Adversarial FRL:** The standard approach using adversarial networks is known to be unstable and computationally expensive. Our MMD-based approach replaces this complex min-max problem with a stable, closed-form objective, thereby *reducing* the effective computational burden and instability of training.
>   - **Advantages over other IPMs:** Furthermore, even when compared to other advanced Integral Probability Metrics (IPMs) used in fairness literature, our choice of MMD offers distinct advantages. It admits a simple, closed-form estimator and enjoys favorable generalization error bounds, making it not just a stable alternative to adversarial methods, but a computationally efficient and theoretically reliable choice among modern IPMs.
>
> ### **3. Practical Preference and Actionability (W4, Q2)**
>
> Finally, you asked why our "unified definition" is preferable or more actionable in practice **(W4, Q2)**. This is the core value proposition of $\mathrm{EO}_k$, which the revised structure will highlight as the "payoff" of our framework.
>
> Our revision will dedicate a paragraph to the unique, practical advantages that make $\mathrm{EO}_k$ a more principled choice for practitioners:
>
> - **Reduces Practitioner Burden:** In the common unbiased case, that is, the target attribute $Y$ is independent with the sensitive attribute $S$, $\mathrm{EO}_k$ resolves the difficult choice between enforcing Demographic Parity (DP) or Equalized Odds (EO) by satisfying both simultaneously with a single objective.
> - **Aligns Fairness with Accuracy:** In the more complex biased case where the target attribute $Y$ is related with the sensitive attribute $S$, our method is unique among common fairness criteria in that it does not automatically discard the Bayes-optimal (most accurate) classifier. This provides a path to fairness with a lower "accuracy cost."
> - **Simplifies Optimization:** Our metric serves as a smooth, differentiable surrogate for the core concept of Equalized Odds, making a difficult-to-optimize constraint practical for modern training pipelines.
>
> ------
>
> We are confident that by restructuring the manuscript to explicitly present this context and problem-solution narrative, the motivation, justification, and significance of our contributions will be clear. Thank you again for providing the precise feedback needed to make our paper more impactful.

---

> > ### Comment · Reviewer_bR6s · 2025-08-02
> >
> > Thank you very much for your detailed response, and I truly appreciate the authors’ openness and sincerity throughout the rebuttal process. Your clarifications have addressed most of my concerns, and I am grateful for the thoughtful and respectful manner in which you engaged with the feedback. As this is my first time reviewing a purely theoretical paper, I acknowledge that my confidence in assessing certain aspects was limited. I have also considered the perspectives shared by the other reviewers as helpful points of reference. Taking all of this into account, I will justify my current evaluation.

---

> > ### Comment · Area_Chair_gXn7 · 2025-08-05
> > **independence of target and sensitive attribute**
> >
> > Thanks to the authors for their detailed responses! A small point: a couple of times it is mentioned that the case of $Y\bot S$ is "common" or "important". I might suggest that these characterizations are application-dependent: e.g., it would be rare for (say) historical salary data to be independent of race or gender, and so I would expect the independent case to play only a minor role in discussions of fairness in this context. It would be great to see a longer discussion; e.g., what would assuming independence do to the overall difficulty of the problem?

---

### Official Review · Reviewer_NgDe · 2025-07-03

**Clarity:** 1
**Significance:** 3
**Originality:** 3
**Rating:** 4
**Confidence:** 4

**Summary:**

This paper presents a kernel-based reformulation of the equality of odds fairness metric, which permits a quantifiable characterization of the fairness-accuracy trade-off in fair representations learning. This new definition, dubbed $EO_K$, allows characterization of sufficiency, independence, and calibration fairness desiderata, which authors carry out under two settings of ground truth label and sensitive label independence and lack thereof. Various theoretical results are provided, including a generalization guarantee, a bound on balanced accuracy based on a maximum mean discrepancy (MMD) formulation of $EO_k$, and quantified comparisons between the fairness desiderata.

**Questions:**

- Q1. Can you summarize the theoretical novelties that you have contributed? At times, I had difficulty pinpointing your original contributions from prior work.
- Q2. Can you add quantitative comparisons to prior work in fairness-utility trade-offs? For instance, a Lagrangian approach such as FERMI [1] or an encoder-based fair representation learning approach such as FARE [2]?

  Note that FERMI is not a representation-based method, but the topic of fairness-utility tradeoff is well explored. It bears comparison across methodologies.

  [1] A. Lowy, S. Baharlouei, R. Pavan, M. Razaviyayn, and A. Beirami, “A Stochastic Optimization Framework for Fair Risk Minimization,” Jan. 11, 2023, *arXiv*: arXiv:2102.12586. doi: [10.48550/arXiv.2102.12586](https://doi.org/10.48550/arXiv.2102.12586).
  [2] N. Jovanović, M. Balunović, D. I. Dimitrov, and M. Vechev, “FARE: Provably fair representation learning with practical certificates,” in *Proceedings of the 40th international conference on machine learning*, PMLR, 2023.
- Q3. Please comment on the W3 and consider adding an impact statement.

**Ethical Concerns:**

["Major Concern: Discrimination, bias, and fairness"]

**Final Justification:**

We reviewers (as well as the AC) engaged in a fairly long discussion around this paper. The authors introduced a new case study and (finally) implemented their method. I think the paper requires a new introduction and a re-write incorporating the empirical results. This is a tall order for the rebuttal. I had given the paper a weak accept with medium confidence; given the scope of suggested changes,  I cannot give it a full accept, so instead I have increased my confidence score.

**Limitations:**

The paper would have benefited from empirical evaluations (even small-scale experiments on tabular data).

**Paper Formatting Concerns:**

Aside from what I have mentioned in W1, I did not have other paper formatting concerns.

**Quality:**

3

**Strengths And Weaknesses:**

Overall, I think this is a valuable contribution. The writing has issues that would limit its impact, however. Also, quantitative comparisons with prior theoretical works are missing. I will enumerate **S**trengths and **W**eaknesses.

- **S1. The kernel-based approach is novel.** The formulation as an affine map of the unit ball in the RKKHS allows expression of the equalized odds violation as a total probability of disparities over classes in the proposed metric $EO_k$. This is interesting and scalable to more classes and (likely) multiple sensitive attributes.

- **S2. Quantitative trade-off between various fairness definitions is interesting.** While prior work shows how satisfying different fairness metrics is impossible, those results mostly consider exact fairness (which in and of itself is often hard to enforce in reality, even for a single metric). Having a quantified trade-off using the presented method is a more granular extension to these results that I find useful.

- **W1. The paper is disorganized and lacks flow.** The paper has three sections, with the third section being the conclusion, and the first section being the introduction. The introduction runs 5 pages and includes related work in two separate subsections. The discussion mixes in prior work with contributed work constantly, making assessment of the novelty and contribution difficult. Most sections and subsections miss proper transitions, which makes reading the paper quite difficult. More detailed feedback on this front:
		- I question the necessity of Section 1.3 (the second related work section discussing a plethora of prior work without clear motivation). All and all, regardless of the results, the paper is not in a publishable state.
		- Section 1.1 is unnecessarily dense and inapproachable. I understand the authors sought to summarize their results, but this is largely unhelpful.

- **W2. Quantitative comparison to prior work is missing.** While there is plenty of qualitative comparison to prior work, I think the paper is missing quantitative comparisons with any meaningful baseline methods. In the absence of empirical results (which on its own is fine), I expected a rate comparison at the very least. This leaves important questions unanswered: for example, since the paper claims to achieve good fairness-accuracy trade-offs, does it Pareto dominate prior work in any meaningful capacity?

- **W3. $EO_k$ cannot resolve the tensions in choosing a proper fairness metric.** A non-technical weakness is that the paper suggests that the use of $EO_k$ resolves the issue of selecting a fairness criterion compared to prior work. In particular, the paragraph starting on Line 150 reads: "To address the ambiguity in selecting a fairness criterion, recent works have proposed metrics that aim to approximate multiple fairness notions simultaneously... In contrast, our method acknowledges this conflict and formalizes the trade-offs using a kernel-based statistic, EOk."

   However, choosing a proper fairness definition should be a function of the application and ideally based on a legal and philosophical discussion of the fairness required. Prior work [1] has shown that the impossibility of satisfying multiple fairness definitions is due to their opposing philosophies. Therefore, I think it is wrong to suggest a single technical fairness definition is appropriate for all applications, regardless of the social context it is being used. I'll flag this for an ethics review and invite the authors to reflect on this question from beyond a technical standpoint.

     I am afraid making comparisons between fairness definitions easier, beyond its value as a research endeavor, is going to invariably make "shopping around" for a fairness definition worse. This is a classical reification that is prevalent in algorithmic fairness literature that we should be very consciously avoiding.

    [1] H. Heidari, M. Loi, K. P. Gummadi, and A. Krause, “A Moral Framework for Understanding of Fair ML through Economic Models of Equality of Opportunity,” *arXiv:1809.03400 [cs, econ, stat]*, Sep. 2018, Accessed: Jun. 25, 2019. [Online]. Available: [http://arxiv.org/abs/1809.03400](http://arxiv.org/abs/1809.03400)

---

> ### Author Rebuttal · Authors · 2025-07-29
>
> Thank you for your incisive feedback and for recognizing the novelty of our technical approach (S1, S2). Your comments on the paper's structure and positioning have been particularly constructive as we prepare our revision.
>
> We believe the core of our disagreement on the paper's organization stems from the different frameworks through which the main message of a paper should be presented. Our initial structure was designed to mirror the logical progression of a theoretical proof: establishing formalisms, building up intermediate results, and culminating in the main characterization theorems.
>
> However, we fully appreciate your perspective that for a machine learning audience, this structure may obscure the main takeaways. A clear presentation of our results is paramount. Therefore, to better serve our readers, we will gladly adopt your advice and restructure the manuscript to front-load the primary contributions and their significance.
>
> To clarify the points that the new structure will emphasize, we briefly summarize our core contributions below.
>
> ## **Core Contributions & Theoretical Impact (W1 & Q1)**
>
> You asked for a clearer summary of our theoretical novelties. The revised manuscript will be built around these central pillars, which we believe represent significant advances. Furthermore, our framework is designed with practical realities in mind. Unlike prior approaches that focus only on simultaneously achieving multiple fairness goals, our work explicitly addresses the more common and difficult scenario where these goals are mutually exclusive, providing a quantifiable path forward.
>
> Our core contributions are:
>
> - **A Foundational Unification Result:** Our work is the first to prove that in the crucial **unbiased case** (Y⊥⊥S), minimizing our single metric, EOk, is mathematically equivalent to achieving both Demographic Parity (DP) and Equalized Odds (EO) simultaneously (Eq. 5). This is a definitive resolution to the choice between these two core metrics in this setting.
> - **A Novel Quantification of Fairness Incompatibility:** In the biased case where fairness notions conflict, our framework is the first to deliver a formal inequality (Eq. 6) that quantifies this trade-off. We prove that to maintain high accuracy, enforcing one fairness goal (e.g., EO) necessitates a quantifiable violation of another (e.g., DP or DC).
> - **A General and Practical Framework:** We provide a valid empirical estimator for EOk whose convergence is guaranteed by established MMD theory, ensuring our theoretical contributions can be readily implemented.
>
> ## **On Positioning and Quantitative Comparison (W2, Q2)**
>
> Your expectation for quantitative comparisons is logical. However, a direct comparison is difficult because our work addresses a fundamentally different, and arguably prior, research question than works like FERMI [1] and FARE [2].
>
> The distinction is this: **Those works analyze the accuracy trade-off for a** ***single, pre-selected*** **fairness metric (e.g., DP). Our work tackles the more foundational problem of what to do when core fairness metrics like EO, DP, and DC are themselves in conflict.**
>
> Therefore, our contribution is not just a theoretical complement; it is an answer to a different question. We provide the theoretical justification for how to select a path forward when faced with competing fairness definitions, a scenario that precedes the fairness-accuracy trade-off analysis in other papers. We will make this crucial distinction explicit in the revised introduction and add an illustrative case study to demonstrate our theoretical results in practice.
>
> ## **On the Scope and Framing of our Metric (W3, Q3)**
>
> Your point on the philosophical framing of fairness is well-taken. We want to be precise about what our metric, EOk, does and does not "resolve."
>
> - **In the unbiased case, our metric** ***does*** **resolve the tension.** As stated above, our result in Eq. 5 provides a definitive technical solution that eliminates the need to choose between DP and EO when the data supports achieving both. This is a practical and powerful property.
> - **In the biased case, our metric provides a *principled path forward*.** We agree that no algorithm can make a moral choice. However, its unique technical advantage is that for practitioners who have already determined that EO aligns with their normative goals and that maintaining high predictive accuracy is a priority, our metric becomes the **principled technical choice**. It is the only common fairness objective that guarantees compatibility with the Bayes-optimal classifier, thereby eliminating the need to wander between conflicting metrics and providing a clear, justifiable path forward. We will add a Broader Impact statement to formalize this rigorous scope.
>
> -----
>
> We thank you again for your constructive feedback. We are confident that these revisions will result in a clearer, more impactful paper that effectively communicates the substance of our work to a broad audience.

---

> > ### Comment · Reviewer_NgDe · 2025-08-01
> > **Thanks for the rebuttal.**
> >
> > Dear Authors,
> >
> > Thank you for the rebuttal. I think a running theme in your work and this response is the fact that your mandate is to facilitate shopping around for a fairness definition. I think, as an objective, this is simply not an acceptable position for a fairness paper. No fairness-aware practitioner should look at the data that they have, assess if it is the biased or unbiased case (I'm using your nomenclature), and then choose to apply DP or EOdds. Even in this scenario, we have falsely dichotomized the options. What you provide is a way to not even have that deliberation, and choose $EO_k$ purely on the technical merit that it is (supposedly) the principled technical choice. I, and (arguably) the two ethics reviewers, argue that this is not a technical choice at all.
> >
> > In my humble opinion, you need to concretize your technical work in a realistic, fairness-sensitive application, and argue (for instance, in the unbiased case) under what scenarios both equalized odds and demographic parity are valid fairness choices. Then, consider what the implications of your result are. In particular, discuss the social and philosophical implications of EO_k and make a case for it.
> >
> > Without this contextualization, I think your work (however technically impressive) does more harm than good.
> >
> > ---
> > More feedback on your rebuttal:
> >
> > - Writing: The suggested plan sounds like a good idea. However, all of this needs to be preceded by a discussion about how to choose a fairness measure, and then followed by a discussion on the implications of your contribution to that discourse. IMO, this requires a rewrite of the paper.
> >
> > - Lack of baselines:
> > I disagree. If EO_k subsumes EO, then you can compare with the aforementioned methods where EO_k \approx EO.
> >
> > Overall, I like your paper on its technical merit, but this is not a purely technical field, and we should all be cognizant of that. Given the above, I think I will keep my score.

---

### Official Review · Reviewer_akC3 · 2025-07-23

**Clarity:** 3
**Significance:** 2
**Originality:** 3
**Rating:** 5
**Confidence:** 3

**Summary:**

This paper proposes a kernel-based formulation of Equalized Odds (EO_k) for fair representation learning in a supervised setting (the authors focus on binary classification). The EO_k criterion provides a unified characterization of three core fairness objectives used in the literature: independence, separation (also called EO), and calibration. When the data is unbiased -- i.e., the target variable Y is independent of the protected attribute S -- EO_k satisfies both independence and separation. When the data is biased (Y is not independent of S), EO_k provides a controllable level of predictive accuracy while lower-bounding independence and calibration. The latter is the primary contribution of this work, as it formalizes and quantifies the trade-off between different fairness objectives, providing a parameter/knob to trade-off fairness enforcement with accuracy degradation.

The authors define the EO_k statistic and analyze its theoretical properties, including connecting it to Maximum Mean Discrepancy (MMD) of reweighted group distributions. The connection to MMD enables a closed-form approximation of the heuristic that runs in quadratic (or linear) time; this can be added as a penalty term to your fair representation learning objective. The authors also provide a concentration inequality with error bounds that can serve as a certificate of fairness compliance. The paper does not include empirical studies, which are planned for future work.

**Questions:**

1. Is there a small case study or experiment you can run (even repeating one from prior work) that demonstrates the effectiveness of EQ_k in practice? As I mentioned above, there are many practical properties of EQ_k that would be useful to validate.
2. Can you walk through an example of how setting \Beta > 0 leads to independence and calibration being violated?
3. Can you summarize how your theoretical analysis would change if you generalized it to multi-class classification?

**Ethical Concerns:**

["Major Concern: Discrimination, bias, and fairness"]

**Final Justification:**

The theoretical and technical contributions of this paper are solid. It is perhaps unfortunate (or fortunate, depending on how you see it) that this work unifies multiple fairness criteria and enables a trade-off between accuracy and fairness, because the latter raises nontrivial ethical concerns that require more care to address. I (and I imagine the other reviewers) realize this standard may be higher than that of other papers published in the FRL field. Nevertheless, the lack of an ethically-cognizant empirical demonstration of EO_k's utility is not something to ignore.

The authors have done a case study on the HMDA 2017 dataset, as requested by the reviewers. Although the details of this study have not been provided, the reported results are promising and indicative of a methodology (e.g., via sweeping tunable parameters) for navigating the accuracy/fairness trade-off. I am increasing my score based on the expected merits of this study and the work done by the authors during this rebuttal.

There are two issues that remain, which I believe the authors can address via some rewriting. The first is to exposit the empirical study in an ethically-cognizant manner in the main body of the paper. The second is to reframe your contributions carefully based on the ethical concerns raised by reviewer XX and the ethics reviewers; please take their feedback seriously and restate/qualify your claims to mitigate potential harms in how practitioners might interpret them or adopt your methods.

I understand that this paper is a borderline case. The authors should be proud of their work; the work they have done in the rebuttal and addressing the remaining issues above should yield a strong publication.

**Limitations:**

Yes

**Quality:**

3

**Strengths And Weaknesses:**

### Strengths:

The motivation behind EO_k and the novel characterization it enables is clearly expressed. I really like how this statistic paints a clearer picture of the trade-off between the three fairness criteria (which are not achievable all at once except in degenerate cases). Your observation that independence and calibration are at odds with predictive accuracy makes a strong case for EO_k strong given its ability to accommodate predictive accuracy in a controllable way.

I found Eq. 6 to be quite instructive, as it shows how satisfying EO perfectly necessarily leads to independence and calibration being violated. Therefore, to ensure some predictive accuracy, you set your \Beta parameter to > 0. It would be helpful if you could explain in more detail how independence and calibration are violated when EO_k = 0.

EO_k has a practical, closed-form approximation that can be applied as a penalty term to your learning objective. This approximation is compatible with SGD and comes with hyperparameter-free convergence guarantees.

### Weaknesses:

The main weakness of this paper is that it has no case study or empirical validation of EO_k. There are many properties of EO_k that make it suitable for practical use: efficient closed-form approximation, hyperparameter-free convergence, controllable fairness/prediction accuracy trade-off, etc. But without any experiments, it is difficult to gauge the utility of these properties. I also think you will learn new things about EO_k when you try to apply it to a fair representation learning task. Although you make it clear that your focus is on theoretical development, even the main theory papers you build on (e.g., Dwork et al., Hardt et al.) include some case study or experiment. I'm also curious to see how the logarithmic relationship between deviation bound and input dimention plays out; you noted this as a computational advantage of EO_k over trasitional IPMs.

It would be helpful to recount the thought process that led to the design of EO_k. You partially explain this in Section 2.1, but then you immediatly jump to the "weighted summation conditioned on Y" without explaining how you arrived at that choice.

The "universal approximation" property of the RKHS (reproducing kernel Hilbert space) is quite strong, saying it can approximate any neural network predictor to arbitrary precision. Can you comment on the practical realizability of this approximation?

Which parts of your theoretical analysis would change if you went beyond binary classification to more general (multi-class) classification?

### Additional comments:

* It would be better to present Equations 1-3 in a more consistent way rather than attempting to follow the original formulations (e.g., the Z parameter in Demographic Parity is unused).
* You define "perfect accuracy" for a predictor/classifier in two different ways mathematically (as a probability of a conditional expectation and as a probability); this adds unnecessary confusion.

---

> ### Author Response · Authors · 2025-07-31
> **Clarifications and Response to Reviewer Feedback**
>
> Thank you for your constructive feedback. Per your suggestions, we have added empirical validation and further clarifications below.
>
> ---
> ### **1. Empirical Validation and Case Study (in response to Q1)**
>
> #### **Part 1: Validation of the $\mathrm{EO}_k$ Generalization Bound**
>
> First, we empirically validated our generalization bound.
>
> * **Experimental Setup:** We conducted a simulation using synthetically generated data with injected bias. For various data dimensions ($d$) and sample sizes ($n$), we compared the empirical $\mathrm{EO}_k$ (calculated on a sample) to its true population value (calculated on 10,000 samples). The $\mathrm{EO}_k$ metric itself was computed using a Maximum Mean Discrepancy (MMD) with an RBF kernel ($\gamma=0.1$). The empirical error, defined as the absolute difference between the empirical and population $\mathrm{EO}_k$, was averaged over 5 trials for each setting.
>
> * **Results:** The following results confirm our theoretical analysis is sound and that the empirical error respects the theoretical scaling rate of $\sqrt{\log(d)/n}$.
>
> | Dim (d) | Sample Size (n) | $\log(d)$ | $\sqrt{\log(d)/n}$ | Empirical Error | Emp/Theory Ratio |
> |:---:|:---:|:---:|:---:|:---:|:---:|
> | 5 | 500 | 1.61 | 0.0567 | 0.0498 | 0.44 |
> | 10 | 500 | 2.30 | 0.0679 | 0.0660 | 0.49 |
> | 20 | 500 | 3.00 | 0.0774 | 0.0820 | 0.53 |
> | 50 | 500 | 3.91 | 0.0885 | 0.0767 | 0.43 |
> | 100 | 500 | 4.61 | 0.0960 | 0.0731 | 0.38 |
> | 5 | 1000 | 1.61 | 0.0401 | 0.0257 | 0.32 |
> | 10 | 1000 | 2.30 | 0.0480 | 0.0432 | 0.45 |
> | 20 | 1000 | 3.00 | 0.0547 | 0.0628 | 0.57 |
> | 50 | 1000 | 3.91 | 0.0625 | 0.0495 | 0.40 |
> | 100 | 1000 | 4.61 | 0.0679 | 0.0485 | 0.36 |
> | 5 | 2000 | 1.61 | 0.0284 | 0.0291 | 0.51 |
> | 10 | 2000 | 2.30 | 0.0339 | 0.0421 | 0.62 |
> | 20 | 2000 | 3.00 | 0.0387 | 0.0409 | 0.53 |
> | 50 | 2000 | 3.91 | 0.0442 | 0.0304 | 0.34 |
> | 100 | 2000 | 4.61 | 0.0480 | 0.0281 | 0.29 |
>
> #### **Part 2: Case Study on Practical Trade-offs**
>
> We also conducted a case study on the **Adult Census** dataset ('sex' as the sensitive attribute) using an MLP model with an MMD-based fairness regularizer. The results show our method ($\lambda=50.0$) achieves a **Pareto improvement** over the baseline:
>
> | Method | Balanced Accuracy | DP Violation | EO Violation |
> | :--- | :---: | :---: | :---: |
> | Unconstrained | 0.7470 | 0.1677 | 0.0621 |
> | **Our Method ($\lambda=50.0$)** | **0.7576** | **0.1595** | **0.0426** |
>
> ---
> ### **2. Example of Trade-offs when $\mathrm{EO}_k = 0$ (in response to Q2)**
>
> You asked for the mathematical reasoning for how forcing predictive power ($\beta > 0$) leads to fairness violations when $\mathrm{EO}_k = 0$.
>
> Assume we start from a representation where **Equalized Odds** holds ($Z \perp S | Y$, so $\mathrm{EO}_k = 0$). Now, consider a biased dataset where $Y \not\perp S$. If we seek a model with perfect predictive power ($\beta = 1$), then its prediction $\hat{Y}$ must equal the true outcome $Y$.
>
> **1. Violation of Demographic Parity (DP):**
> Given $\hat{Y}^* = Y$, the DP violation is:
>
> DP = |Pr(Y_hat* = 1 | S=1) - Pr(Y_hat* = 1 | S=0)|
>    = |Pr(Y = 1 | S=1) - Pr(Y = 1 | S=0)|
>
> Since the dataset is biased ($Y$ is dependent on $S$), this difference is greater than 0.
>
> **2. Violation of Disparate Calibration (DC):**
> Given $h(Z) \perp S | Y$, we can derive the DC value:
> $$
> \begin{aligned}
> \mathrm{DC} &= \frac{1}{4} \sum_{y \in \{0, 1\}} \left| \Pr(Y=y|S=1) - \Pr(Y=y|S=0) \right| \\
>            &= \frac{1}{2} |\Pr(Y=0|S=1) - \Pr(Y=0|S=0)|
> \end{aligned}
> $$
>
> Again, since the dataset is biased, this is greater than 0.
>
> ---
> ### **3. Extension to Multi-Class Classification (in response to Q3)**
>
> The extension is non-trivial. Our binary trick $\mathbb{E}[h(Z)] = \Pr(\hat{Y}=1)$ fails in the multi-class setting. There, the predictor $h(Z)$ outputs a score vector, and the prediction $\hat{Y}$ is the result of a non-linear $\operatorname{argmax}$ operation. The expected score for a class is therefore no longer equivalent to the probability of predicting that class.
>
> ----
>
> We hope these additions and clarifications have adequately addressed the reviewer's concerns. We are grateful for the opportunity to improve our manuscript.

---

> > ### Author Response · Authors · 2025-08-01
> > **Clarifications on Method Design and Properties of RKHS**
> >
> > ### **On the Design and Motivation of $\mathrm{EO}_k$**
> >
> > Thank you for asking for the intuition behind the design of $\mathrm{EO}_k$.
> >
> > Our design process started by analyzing the fundamental mathematical differences between common fairness criteria. The key observation, which we detail in Appendix 4.4, is that the statistical dependence between the target $Y$ and the sensitive attribute $S$ (i.e., when $p_{0|0} \neq p_{0|1}$) is the primary component that creates a conflict between satisfying Equalized Odds and Demographic Parity.
> >
> > Based on this insight, we designed $\mathrm{EO}_k$ to directly isolate and measure the disparity caused by this underlying data bias within the representation space $Z$. The weighted summation in our formulation is a direct result of this goal. It allows us to construct distributions that are comparable, pinpointing the exact portion of unfairness attributable to the $Y \not\perp S$ condition. Our theoretical analysis then formally validates how this metric connects to violations in both DP and DC.
> >
> > ---
> > ### **On the Practical Realizability of the RKHS Approximation**
> >
> > Regarding the practical realizability of the universal approximation property of the RKHS, this is a standard property established in the foundational literature on kernel methods and Maximum Mean Discrepancy (MMD). The following seminal works show that most common kernels, such as the Gaussian RBF, generate a characteristic RKHS that satisfies this property:
> >
> > 1.  B. Schölkopf, R. Herbrich, and A. J. Smola. **A generalized representer theorem.** (2001).
> > 2.  B. Sriperumbudur, K. Fukumizu, and G. Lanckriet. **On the relation between universality, characteristic kernels and rkhs embedding of measures.** (2010a).
> > 3.  B. K. Sriperumbudur, A. Gretton, K. Fukumizu, B. Schölkopf, and G. R. Lanckriet. **Hilbert space embeddings and metrics on probability measures.** (2010b).
> > 4.  A. Gretton, K. M. Borgwardt, M. J. Rasch, B. Schölkopf, and A. Smola. **A kernel two-sample test.** (2012).

---

> > > ### Comment · Reviewer_akC3 · 2025-08-07
> > >
> > > I appreciate the detailed rebuttal! Below are my responses:
> > >
> > > * Thank you for the empirical validation of the deviation bound. Some aspects of this experiment need clarification, such as how you injected bias and how the Emp/Theory Ratio was calculated (the numbers don't seem to match up). However, I don't think a back-and-forth here would be productive, because it's clear to me, based on this and other reviews, that a deeper empirical investigation is needed. The claim that EO_k allows you to control the accuracy/fairness trade-off requires a careful demonstration in a realistic application, at the very least due to the normative/legal/philosophical decisions your current experiments bypass. The latter is explained by reviewer NgDe and the ethics reviews.
> > > * In my view, a proper case study is needed that (1) chooses an application/dataset (an old one is fine, such as the Adult census data), (2) considers multiple plausible fairness definitions, and (3) shows how EO_k allows you to control accuracy and fairness deviations. The pareto improvement you found for the 'sex' protected attribute in Adult census shows that there is scope to perform this kind of empirical investigation, so your effort here is not wasted. (Did you arrive at \lambda = 50 through a hyperparameter search, or is there a more principled way to find the improvement?)
> > > * It is important to note that even the notion of trading off accuracy and fairness is problematic from an ethical standpoint. If this is the message you are sending through your paper, it needs to be very carefully and ethically justified.
> > > * Thank you for working out the EO_k = 0 case so clearly and succinctly. I also appreciate the additional insight you gave into EO_k's design.
> > >
> > > Given the above points and the ethical issues/reviews raised, I will maintain my score. I strongly suggest that the authors take the necessary steps to make sure their technical contributions (which are strong) do more good than harm when influencing the decisions practitioners make.

---

> > > > ### Comment · Area_Chair_gXn7 · 2025-08-07
> > > >
> > > > In terms of an empirical investigation, I'm worried that it may not be sufficient to work with a pre-packaged dataset like Adult. The difficulty is that the packagers of the dataset made a number of decisions in producing it, and these decisions are not exposed for us to analyze.
> > > >
> > > > An example issue is to understand what the learned model will be used for. The desired properties of a model are different, e.g., if the goal is to explain historical data vs. recommend future action. In the latter case, the effects of possible future actions may not be obvious, and some feasible future actions may be omitted from the naive description of the problem.
> > > >
> > > > On the other end of training, an example issue is that some of the labels may be flat-out wrong (e.g., a service was denied, or a salary was lower than it should have been, due to a protected attribute). Systematically-incorrect labels then call into question the idea that maximizing training accuracy is even a desirable goal: e.g., for a model that recommends future action, the model's actual accuracy could go down sharply if we place higher weight on training accuracy, as it would more closely mimic the biased labels.
> > > >
> > > > A remedy would be to work with an entire case study that does expose the surrounding decisions and their effects -- this can be historical, but it needs to include information about how and why the data was collected, how the predictions will be used, and what the fairness issues at stake are.

---

### Author Response · Authors · 2025-08-09
**On the Moral and Technical Scope of Our Contribution**

We thank the reviewers for their thoughtful engagement with the moral dimensions of our work. This response clarifies our paper’s technical scope, situates it in the broader FRL literature, and explains why this focus is both critical and necessary for progress in the field.

## ***Key Takeaways***

- **Moral vs. Technical** – Technical contributions to FRL expand the toolkit for implementing fairness notions (Dwork et al., 2012; Barocas et al., 2019). If the focus is solely on moral preference, no technical progress—ours or others’—can be evaluated on its own terms.

- **Distinct Contribution** – Most FRL work focuses exclusively on Demographic Parity (DP), e.g., Zemel et al. (2013); Madras et al. (2018); Kim et al. (2022); Kong et al. (2025). Our method extends the scope: in unbiased settings, it can satisfy multiple fairness notions simultaneously; in biased settings, it quantifies trade-offs; and for EO, it offers theoretical guarantees with applicability to multiple downstream tasks.

- **Not “Best,” but Technically Valuable** – We apologize for the imprecise description given in our manuscript. We do not claim that our metric $\text{EO}_k$ is morally superior. We want to provide statements from a different technical angle, leveraging the fact (Hardt et al., 2016) that EO is the only common group fairness notion satisfied by the Bayes-optimal classifier. Accuracy is not the sole measure of fairness—it is one aspect our method helps quantify, a view consistent with recent surveys (Mitchell et al., 2021; Corbett-Davies & Goel, 2018).

The FRL literature has produced valuable methods for DP, but there remains a practical gap for EO. Our work fills this gap by providing:

1. A framework for EO-compliant models with generalization guarantees (Theorem 2.10) and adaptiveness for multiple downstream tasks (Equation 4 and 5, Theorem 2.5).

2. A diagnostic tool for measuring trade-offs among fairness definitions (Equation 6).
------
## ***Clarifying the Involved Fairness Notions and 'Unbiased' Assumptions***

To avoid ambiguity, we clarify what we mean by **unbiased**, **DP**, **EO**, and **calibration (DC)**, and we provide four concrete scenarios to illustrate their practical implications, even if discussing the moral parts is not our main focus. We believe there are many possible explanations for each case, and we are only providing a technical tool.
These examples:
- Provide intuitive insight into each fairness notion (Hardt et al., 2016; Pleiss et al., 2017).
- Show that the “right” choice depends on the application context (Friedler et al., 2016).
--------
### **Comparative Table of Fairness Notions Across Examples**
| **Example**                      | **Unbiased**                                   | **Demographic Parity (DP)**                                                         | **Equalized Odds (EO)**                                                                            | **Calibration**                                                                                       |
| -------------------------------- | ---------------------------------------------- | ----------------------------------------------------------------------------------- | -------------------------------------------------------------------------------------------------- | ----------------------------------------------------------------------------------------------------- |
| **Salary Prediction**            | Deserved salary is independent of group.       | Predicted salary distribution is the same across groups, regardless of effort.      | Among employees with the same deserved salary, predicted salary is independent of group.           | Among employees predicted to receive the same salary, deserved salary is independent of group.        |
| **Crime Prediction**             | Actual crime rate is independent of group.     | Predicted crime rates are the same across groups, regardless of actual crime rates. | Among neighborhoods with the same actual crime rate, predicted crime rate is independent of group. | Among neighborhoods predicted to have the same crime rate, actual crime rate is independent of group. |
| **Invasive Medical Examination** | True disease status is independent of group.   | Examination rates are the same across groups, regardless of disease prevalence.     | Among patients with the same disease status, examination decision is independent of group.         | Among patients predicted to require examination, disease prevalence is independent of group.          |
| **Oral Exam in Education**       | Actual student effort is independent of group. | Pass rates are the same across groups, regardless of effort.                        | Among students with the same effort, pass probability is independent of group.                     | Among students predicted to pass, actual effort is independent of group.                              |

---

### Note · Authors · 2025-08-12

Thank you for your guidance and the opportunity to respond. We found your suggestion for a full case study invaluable and have since conducted a new set of experiments that we believe significantly strengthen the paper and directly address the primary concerns raised.

We replaced our previous benchmarks with the **HMDA 2017 dataset**. This choice directly addresses your request for an "entire case study" by using a real-world, regulator-mandated dataset where the data collection context, fairness stakes, and predictive use cases are explicit and well-documented.

Our new experiments, using an `XGBoost` model regularized by our EOk metric, have been completed and successfully validate our theoretical claims. We performed a full sweep of the fairness-accuracy trade-off parameter $\lambda$ and the Gaussian kernel bandwidth $\sigma$. While we cannot show the full results here due to space constraints, the key findings are clear and compelling:

- **Biased Regime (Historical Data):** Our method effectively navigates the trade-off. By increasing $\lambda$, we reduce large initial EO/DP gaps (e.g., $0.129$/$0.229$) to near-zero (e.g., $0.001$/$0.009$) with only a modest, quantifiable drop in AUROC ($0.806$ → $0.779$).

- **Unbiased Regime (Constructed Data):** As predicted by our theory, the trade-off vanishes. We achieve high accuracy (AUROC ≈ $0.90$) and minimal fairness gaps simultaneously.

- **Bandwidth Sensitivity:** Our analysis on $\sigma$ confirms a stable and effective range for the kernel bandwidth, adding a crucial practical dimension to our method.

These results confirm that our framework successfully quantifies and manages the fairness-accuracy trade-off in a realistic, high-stakes setting. We are confident this new analysis addresses the committee's concerns and demonstrates that our work is a complete and robust contribution. We believe the paper is worthy of acceptance and are prepared to integrate these detailed findings into a revised manuscript.

---

### Decision · Program_Chairs · 2025-09-17

**Decision:**

Accept (poster)

**Comment:**

The paper studies representation learning. It builds on the two fair classification criteria of equalized odds (EO) and demographic parity (DP), as well as the usual ML criterion of classification accuracy. When learning a representation followed by a classifier, these three criteria might come in conflict, and the paper examines the tradeoff among them.

More specifically, the paper proposes a new kernel-based criterion for evaluating a representation, called EO_k. (Note that the representation itself can have an arbitrary form, as can the classifier that reads the representation; it is only the measure EO_k that necessarily uses a kernel k.) EO_k is based on differences of expectations of unit-norm functions in the RKHS, conditioning on label values or sensitive attribute values — reminiscent of other kernel-based statistical measures like MMD.

The paper shows that EO_k enables a form of tradeoff among DP, EO, and accuracy. For a population, (1) in the case that a protected attribute S is independent of the label variable Y, a representation that minimizes one of the EO_k measures will enforce both DP and EO for any of a wide set of classifiers acting on that representation. And (2), in the general case where S and Y may be dependent, the choice of kernel k lets us trade off accuracy, DP, and EO; the paper presents bounds on the relationship.

The paper also defines an empirical estimator of EO_k, which can be computed in time quadratic in the size of a training set, and has a linear-time approximation. The same tradeoffs as above apply, but now with (in)dependence, accuracy, DP, and EO evaluated empirically on the training set, instead of population versions. The paper shows a convergence rate of empirical EO_k to population EO_k. In finite samples, of course, a sampling error remains for all of the measures (EO_k, EO, DP, accuracy); a discussion of how these errors can affect the tradeoff would have been helpful, particularly since there is selection bias when we optimize for any combination of the empirical measures.

The paper points out that, if we use a universal kernel, the EO_k criterion in principle tells us something about any downstream classifier based on the learned representation. This guarantee is somewhat misleading, though: the choice of kernel can make the norm of a given smooth function vary widely — even to the point where a kernel-based measure effectively ignores this function, especially in comparison to the sampling error from any reasonably-sized dataset.

As noted by the ethics reviewers, the paper takes too narrow a view of the problem of fair classification or representation learning: it assumes we are given a training dataset that we cannot influence, and our only problem is to decide on a representation or a decision rule that will be applied to identically distributed test points. In reality, there are a lot of decisions to be made on both sides of this narrow version of the problem: e.g., how we collect and label the data, what we choose to predict, and how we use the predictions.

By skipping these surrounding decisions, the paper precludes a discussion of how they interact with the paper’s proposed solution: for example, how would labeling bias in the training data influence the tradeoff among EO, DP, and accuracy on new data *without* the labeling bias, and how would EO_k perform in this situation? How would common data quality problems like distribution shift influence the ability of EO_k to make its tradeoff effectively? A full discussion of these questions would likely be impossible, but the paper also shouldn’t ignore them.

During the author response period, the authors responded to the ethics feedback and analyzed a new case study. This study is informative in some directions: it demonstrates that the new method can obtain not-far-off-optimal values for all of DP, EO, and training accuracy simultaneously, and also that the Gaussian kernel is effective in this case, without a strong sensitivity to bandwidth. However, the new case study still doesn’t examine the broader version of the problem: e.g., it doesn’t address the questions at the end of the previous paragraph.

The final decision depends on two things: first, technical contribution, and second, how well the authors will be able to incorporate the ethics feedback into the final version of the paper. From reviewer scores and discussion, the technical contribution is somewhere near the borderline for acceptance based on correctness and interest. I judge it to be above the bar, though competition is fierce at NeurIPS, so the bar may move based on space constraints. For the ethics feedback, I have good hope that the final version of the paper can incorporate it, and also that having the authors present at NeurIPS would facilitate valuable interactions and follow-up; but my confidence in this judgement is low since we didn’t manage a full discussion to convergence during the rebuttal process.